# A discrete subtype of neural progenitor crucial for cortical folding in the gyrencephalic mammalian brain

Naoyuki Matsumoto[1], Satoshi Tanaka[1,2], Toshihide Horiike[1], Yohei Shinmyo[1], Hiroshi Kawasaki[1]*

[1]Department of Medical Neuroscience, Graduate School of Medical Sciences, Kanazawa University, Kanazawa, Japan; [2]Medical Research Training Program, School of Medicine, Kanazawa University, Kanazawa, Japan

**Abstract** An increase in the diversity of neural progenitor subtypes and folding of the cerebral cortex are characteristic features which appeared during the evolution of the mammalian brain. Here, we show that the expansion of a specific subtype of neural progenitor is crucial for cortical folding. We found that outer radial glial (oRG) cells can be subdivided by HOPX expression in the gyrencephalic cerebral cortex of ferrets. Compared with HOPX-negative oRG cells, HOPX-positive oRG cells had high self-renewal activity and were accumulated in prospective gyral regions. Using our in vivo genetic manipulation technique for ferrets, we found that the number of HOPX-positive oRG cells and their self-renewal activity were regulated by sonic hedgehog (Shh) signaling. Importantly, suppressing Shh signaling reduced HOPX-positive oRG cells and cortical folding, while enhancing it had opposing effects. Our results reveal a novel subtype of neural progenitor important for cortical folding in gyrencephalic mammalian cerebral cortex.

*For correspondence:
hiroshi-kawasaki@umin.ac.jp

**Competing interests:** The authors declare that no competing interests exist.

## Introduction

Crucial advances of the mammalian brain during evolution include the expansion and folding of the cerebral cortex (*Molnár et al., 2006*; *Rakic, 2009*). The brains of humans, monkeys and ferrets have a folded cerebral cortex (i.e. gyrencephalic cortex), whereas those of rodents often have a smooth cerebral cortex (lissencephalic cortex). Cortical folding is believed to underlie the acquisition of higher brain functions during development and evolution (*Florio and Huttner, 2014*; *Llinares-Benadero and Borrell, 2019*; *Lui et al., 2011*; *Sun and Hevner, 2014*). Consistent with this proposition, malformations of cortical folds are often associated with intellectual disabilities, epilepsy and diseases such as schizophrenia and autism (*Fernández et al., 2016*; *Ross and Walsh, 2001*). Thus, uncovering the molecular and cellular mechanisms that underlie the development and malformation of cortical folds has been an important issue. However, our understanding of the mechanisms of cortical folding is still rudimentary.

An increase in the diversity of neural progenitor subtypes is associated with the appearance of cortical folding during evolution (*Dehay et al., 2015*; *Fernández et al., 2016*; *Florio and Huttner, 2014*; *Kriegstein et al., 2006*; *Molnár et al., 2006*; *Rakic, 2009*; *Sun and Hevner, 2014*). The developing cerebral cortex of the lissencephalic mouse brain contains two germinal layers, the ventricular zone (VZ) and the subventricular zone (SVZ), which comprise radial glial cells (RG cells, also known as apical progenitors/ventricular RG cells/apical RG cells) and intermediate progenitor (IP) cells, respectively. The SVZ of humans, monkeys and ferrets is radially expanded and further subdivided into the inner SVZ (ISVZ) and the outer SVZ (OSVZ), which contains abundant outer radial glial cells (oRG cells, also known as OSVZ RG cells/basal RG cells/intermediate RG cells/translocating RG cells). Although it has been proposed that the acquisition of cortical folds during evolution resulted

from increased numbers and diversity of neural progenitors (*Dehay et al., 2015*; *Fernández et al., 2016*; *Florio and Huttner, 2014*; *Kriegstein et al., 2006*; *Molnár et al., 2006*; *Rakic, 2009*; *Sun and Hevner, 2014*), it has been difficult to test this proposal experimentally in vivo. This is at least partially because of a paucity of rapid and efficient genetic manipulation techniques that could be applied to gyrencephalic brains. To overcome these difficulties, we recently developed genetic manipulation techniques for ferrets using in utero electroporation (IUE) and the CRISPR/Cas9 system (*Kawasaki et al., 2012*; *Kawasaki et al., 2013*; *Shinmyo et al., 2017*). Using these techniques, we showed that reductions in SVZ progenitors resulted in the impairment of cortical folding in the naturally gyrencephalic ferret in vivo (*Matsumoto et al., 2017b*; *Toda et al., 2016*). Although these data indicate the importance of SVZ progenitors for cortical folding, increased SVZ progenitors in the lissencephalic mouse cerebral cortex was not sufficient for inducing de novo formation of cortical folds (*Nonaka-Kinoshita et al., 2013*). Therefore, the expansion of specific subtypes of SVZ progenitors which are found only in the gyrencephalic brains seemed to be crucial for the formation of cortical folds. However, the subtypes of SVZ progenitors responsible for cortical folding were still unclear.

Homeodomain-only protein homeobox (HOPX) was recently identified as a marker of oRG cells in the developing human cerebral cortex (*Pollen et al., 2015*). HOPX was also reported to be expressed in oRG cells of other gyrencephalic animals such as monkeys and ferrets (*Johnson et al., 2018*; *Pollen et al., 2015*). Interestingly, a previous study showed that not all oRG cells express HOPX in the human cerebral cortex (*Thomsen et al., 2016*), raising the possibility that oRG cells consist of multiple subtypes of neural progenitors. Here, we show that oRG cells can be divided into two groups: HOPX-positive and HOPX-negative. HOPX-positive oRG cells were preferentially distributed in prospective gyri and exhibited high self-renewal activity. Using our IUE techniques for ferrets, we found that sonic hedgehog (Shh) signaling suppressed the differentiation of HOPX-positive oRG cells and, as a result, increased the number of HOPX-positive oRG cells. Furthermore, reducing the number of HOPX-positive oRG cells by suppressing Shh signaling impaired cortical folding, while an increase in HOPX-positive oRG cells induced by the activation of Shh signaling resulted in additional cortical folds. Our results reveal a novel subtype of neural progenitor important for cortical folding in the gyrencephalic mammalian cerebral cortex.

## Results

### HOPX-positive and HOPX-negative oRG cells in the developing ferret cerebral cortex have distinct cellular properties

To investigate the subtypes of neural progenitors in the OSVZ, we performed triple immunostaining for Pax6, Tbr2 and HOPX using the developing ferret cerebral cortex at P1 (*Figure 1A,B*). It was reported that oRG cells in the OSVZ are Pax6-positive and Tbr2-negative, and that IP cells express Tbr2 (*Fietz et al., 2010*; *Hansen et al., 2010*; *Reillo et al., 2011*). HOPX was recently identified as a marker of human oRG cells (*Pollen et al., 2015*). Consistent with this, almost all HOPX-positive cells in the OSVZ expressed Pax6 (96.5 ± 2.1%) but rarely expressed Tbr2 (7.2 ± 2.0%), suggesting that HOPX-positive cells in the OSVZ are oRG cells in the ferret cerebral cortex (*Figure 1A,B*).

We noticed that HOPX was expressed in a subset of oRG cells (*Figure 1B*), raising the possibility that oRG cells can be divided into two subtypes: HOPX-positive and HOPX-negative. We next examined the detailed distribution patterns of HOPX-positive oRG cells in the developing ferret cerebral cortex. Interestingly, the abundance of HOPX-positive oRG cells in the OSVZ had regional differences along the anteroposterior (AP) axis (*Figure 1C*). To determine whether the abundance of HOPX-positive oRG cells correlated with the positions of prospective gyri and sulci, five regions in the germinal zones were selected along the AP axis in accordance with the positions of prospective gyri (regions #1, #3 and #5 in *Figure 1C*) and sulci (region #2 in *Figure 1C*), as reported previously (*de Juan Romero et al., 2015*). We found that HOPX-positive oRG cells were fewer in region #2 than in regions #1, #3 and #5 (*Figure 1C*). Importantly, the difference in the number of HOPX-positive oRG cells between regions #3 and #2 was much greater than that of Pax6-positive cells between the same regions (*Figure 1C–E*), suggesting that HOPX-positive oRG cells, rather than HOPX-negative oRG cells, preferentially accumulate in prospective gyral regions.

To confirm this idea, we counted the numbers of HOPX-positive oRG cells (Pax6+/Tbr2-/HOPX+), HOPX-negative oRG cells (Pax6+/Tbr2-/HOPX-), IP cells (Tbr2+), Pax6-positive cells and all

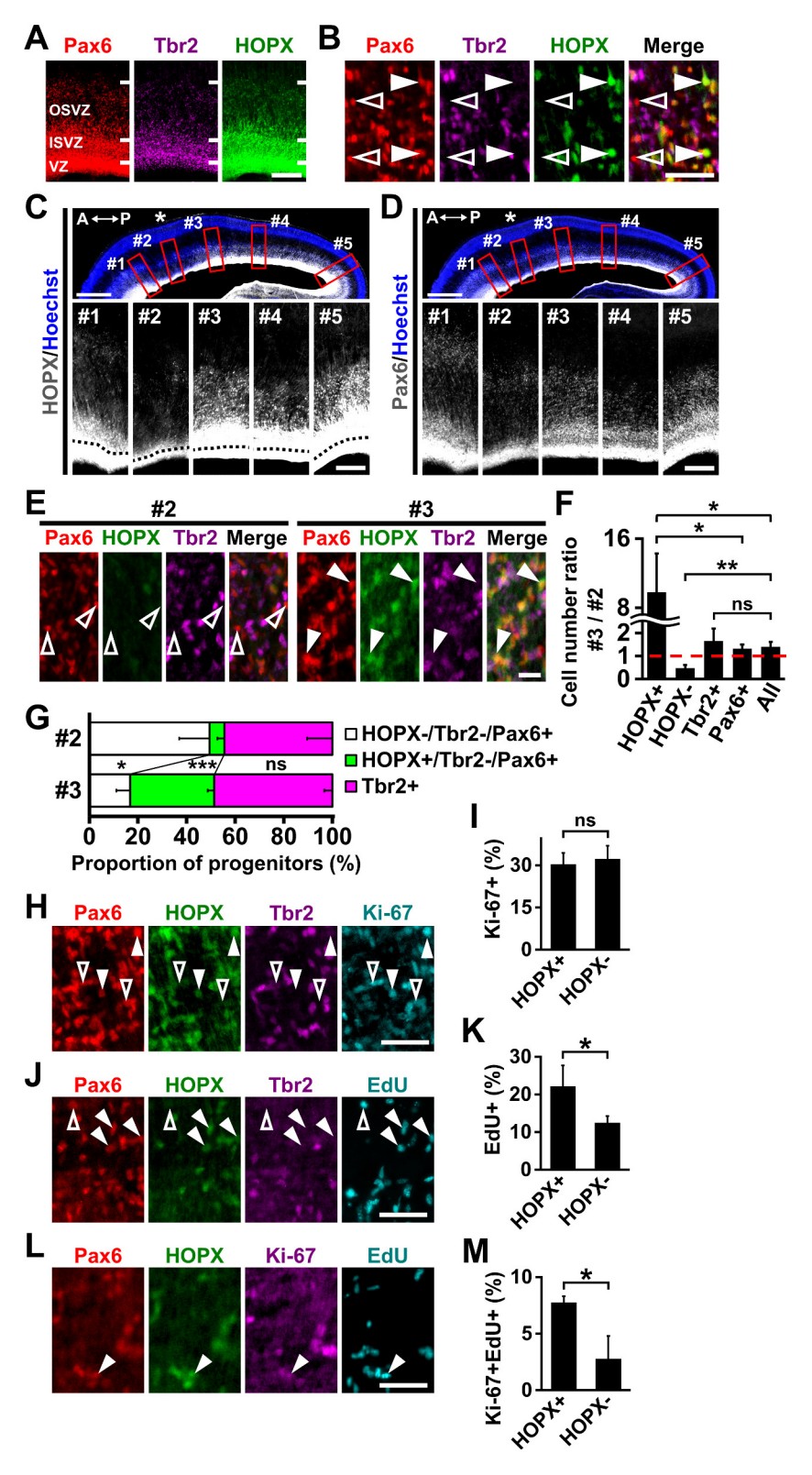

**Figure 1.** HOPX-positive oRG cells are preferentially distributed in prospective gyral regions in the developing ferret cerebral cortex. (**A, B**) Sections of the ferret cerebral cortex at P1 were subjected to immunohistochemistry for Pax6, Tbr2 and HOPX. (**B**) Higher magnification images of the OSVZ are shown. HOPX was expressed in a subset of oRG cells (Pax6-positive and Tbr2-negative). Arrowheads and open arrowheads indicate HOPX-positive and HOPX-negative oRG cells, respectively. Scale bars = 200 μm (**A**), 50 μm (**B**). (**C, D**) Sagittal sections of the ferret brain at P1 were subjected to Hoechst

*Figure 1 continued on next page*

*Figure 1 continued*

33342 staining plus immunohistochemistry for HOPX (**C**) and Pax6 (**D**). Asterisks indicate prospective sulcal regions. Five regions (red boxes, #1–#5) based on the positions of prospective gyri and sulci in the upper panels are magnified in the lower panels. Black broken lines indicate the border between the VZ and the ISVZ. A, anterior; P, posterior. Scale bars = 1 mm (upper), 200 µm (lower). (**E**) Sections of the ferret cerebral cortex at P1 were subjected to immunohistochemistry for Pax6, HOPX and Tbr2. Magnified images of the OSVZ corresponding to regions #2 and #3 are shown. Arrowheads and open arrowheads indicate HOPX-positive and HOPX-negative oRG cells, respectively. Scale bar = 20 µm. (**F**) Ratios of cell numbers between prospective gyral and sulcal regions. Numbers of HOPX-positive oRG cells (HOPX+), HOPX-negative oRG cells (HOPX-), IP cells (Tbr2+), Pax6-positive cells and Tbr2-positive and/or Pax6-positive cells (All) in region #3 were divided by the corresponding numbers in region #2. The ratio would be one if the number of cells was the same between regions #3 and #2 (broken red line). n = 3 animals. Bars present mean ± SD. ns, not significant. *p<0.05, **p<0.01, Student's *t*-test. (**G**) Proportions of HOPX-positive oRG cells (green), HOPX-negative oRG cells (white) and IP cells (magenta) in regions #2 and #3. n = 3 animals. Bars present mean ± SD. *p<0.05, ***p<0.001, Student's *t*-test. (**H**) Sections of the ferret cerebral cortex at P1 were quadruple-stained with anti-HOPX, anti-Tbr2, anti-Pax6 and anti-Ki-67 antibodies. Arrowheads and open arrowheads indicate HOPX-positive oRG cells co-expressing Ki-67 and HOPX-negative oRG cells co-expressing Ki-67, respectively. Scale bar = 50 µm. (**I**) Percentages of HOPX-positive (HOPX+) and HOPX-negative (HOPX-) oRG cells co-expressing Ki-67 in the OSVZ. n = 3 animals. Bars present mean ± SD. ns, not significant. Student's *t*-test. (**J**) Newborn ferrets were intraperitoneally injected with EdU at P0 and analyzed 28 hr later. Sections were triple-stained with anti-HOPX, anti-Tbr2 and anti-Pax6 antibodies, and EdU was visualized. Arrowheads and open arrowheads indicate HOPX-positive and HOPX-negative oRG cells co-labeled with EdU, respectively. Scale bar = 50 µm. (**K**) Percentages of HOPX-positive (HOPX+) and HOPX-negative (HOPX-) oRG cells co-labeled with EdU in the OSVZ. n = 3 animals for each condition. Bars present mean ± SD. *p<0.05, Student's *t*-test. (**L**) Newborn ferrets were intraperitoneally injected with EdU at P0 and analyzed 28 hr later. Sections were triple-stained with anti-HOPX, anti-Pax6 and anti-Ki-67 antibodies, and EdU was visualized. Arrowhead indicates HOPX-positive oRG cells co-labeled with Ki-67 and EdU. Scale bar = 50 µm. (**M**) Percentages of HOPX-positive (HOPX+) and HOPX-negative (HOPX-) oRG cells co-labeled with Ki-67 and EdU in the OSVZ. n = 3 animals for each condition. Bars present mean ± SD. *p<0.05, Student's *t*-test.

The online version of this article includes the following figure supplement(s) for figure 1:

**Figure supplement 1.** HOPX-positive oRG cells are preferentially distributed in prospective gyral regions in the developing ferret cerebral cortex.

progenitors (Pax6+ and/or Tbr2+) in regions #3 and #2 of the OSVZ and calculated the ratio of cell number in region #3 to that in region #2 (*Figure 1F*). The ratio for HOPX-positive oRG cells was markedly higher than that for all OSVZ progenitors (HOPX+, 9.8 ± 4.4; all, 1.4 ± 0.2; p=0.03; Student's *t*-test) and Pax6-positive cells (HOPX+, 9.8 ± 4.4; Pax6+, 1.3 ± 0.2; p=0.03; Student's *t*-test) (*Figure 1F*), whereas the ratio for HOPX-negative oRG cells was significantly lower than that for all OSVZ progenitors (HOPX-, 0.5 ± 0.1; all, 1.4 ± 0.2; p=0.003; Student's *t*-test) (*Figure 1F*). The ratio for IP cells was comparable to that for all OSVZ progenitors (Tbr2+, 1.7 ± 0.5; all, 1.4 ± 0.2; p=0.29; Student's *t*-test) (*Figure 1F*). Consistently, cell numbers in another prospective gyral region, region #5, were compared with those in region #2, and the ratio of HOPX-positive oRG cells in region #5 to those in region #2 was higher than that for all OSVZ progenitors (HOPX+, 19.9 ± 7.6; all, 1.5 ± 0.2; p=0.01; Student's *t*-test), while the ratio for HOPX-negative oRG cells was lower than that for all OSVZ progenitors (HOPX-, 0.34 ± 0.02; all, 1.5 ± 0.2; p=0.0005; Student's *t*-test) (*Figure 1—figure supplement 1A*). These results indicate that HOPX-positive oRG cells, rather than HOPX-negative oRG cells and IP cells, preferentially accumulate in prospective gyral regions.

We also compared the proportions of progenitor cell types in the OSVZ between prospective gyri and sulci. The percentages of HOPX-positive oRG cells (HOPX+/Tbr2-/Pax6+) in region #3 and #5 were markedly higher than that in region #2 (region #3, 34.6 ± 2.7; region #2, 6.1 ± 2.9; p=0.0002; Student's *t*-test) (region #5, 34.5 ± 1.6; region #2, 3.2 ± 1.7; p=0.00002; Student's *t*-test) (*Figure 1G*, *Figure 1—figure supplement 1B*, green). In contrast, the percentages of HOPX-negative oRG cells (HOPX-/Tbr2-/Pax6+) in regions #3 and #5 were significantly lower than that in region #2 (region #3, 16.8 ± 5.7; region #2, 49.4 ± 12.5; p=0.01; Student's *t*-test) (region #5, 12.1 ± 1.8; region #2, 52.3 ± 3.8; p=0.00009; Student's *t*-test) (*Figure 1G*, *Figure 1—figure supplement 1B*, white). The percentage of IP cells (Tbr2+) in region #3 was comparable with that in region #2 (region #3, 48.6 ± 3.3; region #2, 44.4 ± 10.4; p=0.31; Student's *t*-test) (*Figure 1G*, magenta), and the percentage of IP cells in region #5 was slightly higher than that in region #2 (region #5, 53.4 ± 0.1; region #2, 44.5 ± 5.4; p=0.04; Student's *t*-test) (*Figure 1—figure supplement 1B*). Thus, these findings provide further evidence that HOPX-positive oRG cells selectively accumulate in prospective gyral regions of the ferret cerebral cortex.

Our finding that HOPX-positive and HOPX-negative oRG cells have distinct distribution patterns implied that they also have distinct cellular properties. We therefore examined cell proliferation in these oRG cell types by quadruple immunostaining for HOPX, Tbr2, Pax6 and the cell proliferation marker Ki-67. The percentages of HOPX-positive and -negative oRG cells co-expressing Ki-67 were

comparable (HOPX+, 30.3 ± 4.1; HOPX-, 32.3 ± 4.6; p=0.34; Student's *t*-test) (*Figure 1H,I*), suggesting that the proliferation rate of HOXP-positive oRG cells is similar to that of HOPX-negative oRG cells. We next investigated the percentage of self-renewed oRG cells by injecting 5-ethynyl-2'-deoxyuridine (EdU) at P0 (*Sawada, 2019*). EdU staining was performed using sections prepared 28 hr after the injection because oRG cells undergo cell division within 28 hr after entering S phase in the ferret cerebral cortex (*Turrero García et al., 2016*). We found that the percentage of HOPX-positive oRG cells co-labeled with EdU was significantly higher than that of HOPX-negative oRG cells (HOPX +, 22.2 ± 5.5; HOPX-, 12.4 ± 1.8; p=0.04; Student's *t*-test) (*Figure 1J,K*). Furthermore, to examine whether EdU-positive cells are still in the proliferative state, we performed Ki-67 immunohistochemistry using sections prepared 28 hr after EdU injection. The percentage of HOPX-positive oRG cells co-labeled with Ki-67 and EdU (*Figure 1L*, arrowhead) was significantly higher than that of HOPX-negative oRG cells co-labeled with Ki-67 and EdU (HOPX+, 7.8 ± 0.6; HOPX-, 2.8 ± 2.0; p=0.01; Student's *t*-test) (*Figure 1M*). These results indicate that HOPX-positive oRG cells tend to remain in their progenitor state even after cell division and do not differentiate into the next stage. It seems likely that HOPX-positive and HOPX-negative oRG cells indeed have distinct cellular properties.

## Shh signaling is activated preferentially in HOPX-positive oRG cells in the OSVZ

We next investigated the molecular mechanisms regulating the development of HOPX-positive oRG cells. We found that *GLI1*, whose expression levels are upregulated by Shh signaling (*Lee et al., 1997*), was expressed in the OSVZ of the ferret cerebral cortex. Interestingly, *GLI1* signals in the OSVZ were more abundant in prospective gyri (*Figure 2A*, #1, #3 and #5) than in prospective sulci (*Figure 2A*, #2 and #4). To test in which cell type *GLI1* was expressed, we performed in situ hybridization for *GLI1* and immunostaining for Pax6, Tbr2 and HOPX. We found that *GLI1*-positive cells in the OSVZ were mostly Pax6-positive and Tbr2-negative (Pax6+/Tbr2-, 65.2 ± 9.1%; Tbr2+, 8.0 ± 4.5%; Pax6-/Tbr2-, 26.8 ± 5.0%) (*Figure 2B,D*), suggesting that most of the *GLI1*-positive cells in the OSVZ are oRG cells. Interestingly, *GLI1*-positive oRG cells were mainly HOPX-positive (HOPX+, 92.3 ± 5.5%; HOPX-, 7.7 ± 5.5%) (*Figure 2C,E*). These results indicate that Shh signaling is preferentially activated in HOPX-positive oRG cells rather than in HOPX-negative oRG cells and IP cells.

## Shh signaling enhances the self-renewal of HOPX-positive oRG cells and suppresses their differentiation

Because HOPX-positive oRG cells exhibit lower differentiation rates and higher Shh signaling activity, we hypothesized that Shh signaling suppresses the differentiation of HOXP-positive oRG cells and promotes their self-renewal. To test this, we utilized an untethered form of hedgehog-interacting protein (Hhip) lacking the C-terminal membrane-anchoring domain (HhipΔC22). HhipΔC22 is released from transfected cells and competitively inhibits the binding of Shh to its receptor (*Chuang and McMahon, 1999*; *Kwong et al., 2014*; *Yoshino et al., 2016*). HhipΔC22 therefore suppresses Shh signaling not only in transfected cells but also in neighboring non-transfected cells non-cell-autonomously. We introduced HhipΔC22 into the ferret cerebral cortex using our IUE technique for ferrets (*Kawasaki et al., 2012*; *Kawasaki et al., 2013*) and performed in situ hybridization for *GLI1*. *GLI1* signals were markedly reduced by HhipΔC22 (*Figure 2—figure supplement 1*), indicating that HhipΔC22 strongly suppresses Shh signaling in the ferret cerebral cortex.

To examine the effect of HhipΔC22 on the self-renewal of HOPX-positive oRG cells, we introduced HhipΔC22 into the ferret cerebral cortex using IUE at E33, when electroporation mainly transfects layer 5 neurons (*Figure 2—figure supplement 2*). We then injected EdU at P0 and performed EdU staining on sections obtained 28 hr after the injection. The percentage of HOPX-positive oRG cells co-labeled with EdU was significantly decreased by HhipΔC22 (control, 23.6 ± 4.8; HhipΔC22, 14.2 ± 1.2; p=0.03; Student's *t*-test) (*Figure 2F,G*). In contrast, the percentage of IP cells co-labeled with EdU and that of HOPX-negative oRG cells co-labeled with EdU were not affected by HhipΔC22 (IP cells: control, 32.7 ± 7.9; HhipΔC22, 26.7 ± 2.0; p=0.17; Student's *t*-test) (HOPX-negative oRG cells: control, 11.4 ± 3.8; HhipΔC22, 12.6 ± 1.7; p=0.35; Student's *t*-test) (*Figure 2H–J*). These results suggest that Shh signaling selectively regulates the self-renewal of HOXP-positive oRG cells in the ferret cerebral cortex.

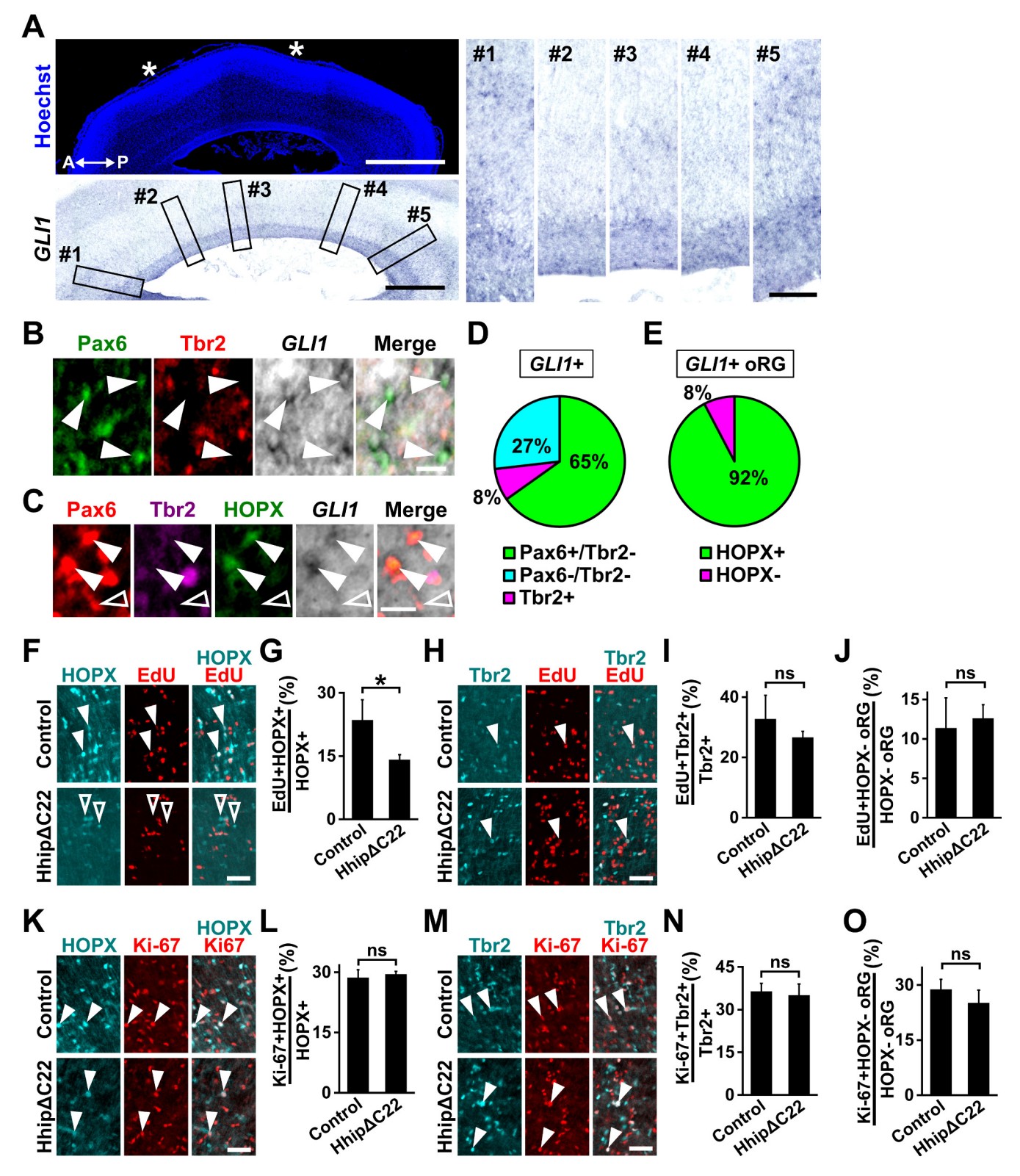

**Figure 2.** Shh signaling is highly activated in HOPX-positive oRG cells and prevents HOPX-positive oRG cells from differentiating. (**A**) Sagittal sections of the ferret brain at P6 were subjected to in situ hybridization for *GLI1* and Hoechst 33342 staining. Asterisks indicate areas of prospective sulci. A higher-magnification image of the germinal zone is shown in the lower panel. Five regions (boxes, #1–#5) based on the positions of prospective gyri and sulci in the left panels were magnified and are shown in the right panels. *GLI1* was more abundantly expressed in the OSVZ of prospective gyri (#1,

*Figure 2 continued on next page*

*Figure 2 continued*

3, 5) than in that of prospective sulci (#2, 4). A, anterior; P, posterior.Scale bars = 2 mm (left, upper), 1 mm (left, lower), 200 µm (right). (B) Sections of the ferret cerebral cortex at P1 were subjected to in situ hybridization for *GLI1* and immunohistochemistry for Pax6 and Tbr2. High-magnification images of the OSVZ are shown. *GLI1* was mainly expressed in oRG cells (Pax6-positive and Tbr2-negative, arrowheads). Scale bar = 20 µm. (C) Sections of the ferret cerebral cortex at P1 were subjected to in situ hybridization for *GLI1* and immunohistochemistry for Pax6, Tbr2 and HOPX. High-magnification images of the OSVZ are shown. *GLI1*-positive oRG cells were mainly HOPX-positive (arrowheads), rather than HOPX-negative (open arrowhead). Scale bar = 20 µm. (D) Percentage of *GLI1*-positive cells co-expressing Pax6 and/or Tbr2. n = 3 animals. (E) Percentage of *GLI1*-positive oRG cells that were also positive for HOPX. n = 3 animals. (F–J) pCAG-EGFP plus either pCAG-HhipΔC22 or pCAG control vector was electroporated into the ferret cerebral cortex at E33. The newborn ferrets were then intraperitoneally injected with EdU at P0 and analyzed 28 hr later. (F) Sections were stained with anti-HOPX antibody, and EdU was visualized. Magnified images of the OSVZ are shown. Arrowheads and open arrowheads indicate HOPX-positive cells co-labeled with and without EdU, respectively. (G) Percentage of HOPX-positive cells co-labeled with EdU in the OSVZ. (H) Sections were stained with anti-Tbr2 antibody, and EdU was visualized. Magnified images of the OSVZ are shown. Arrowheads indicate Tbr2-positive cells co-labeled with EdU. (I) Percentage of Tbr2-positive cells co-labeled with EdU. n = 3 animals for each condition. (J) Percentage of HOPX-negative oRG cells co-labeled with EdU in the OSVZ. Bars present mean ± SD. *p<0.05; ns, not significant; Student's *t*-test. Scale bars = 50 µm. (K–O) pCAG-EGFP plus either pCAG-HhipΔC22 or pCAG control vector was electroporated at E33, and the brains were dissected at P1. (K) Sections were double-stained with anti-HOPX and anti-Ki-67 antibodies, and magnified images of the OSVZ are shown. Arrowheads indicate HOPX-positive cells co-expressing Ki-67. (L) Percentages of HOPX-positive cells co-expressing Ki-67 in the OSVZ. (M) Sections were double-stained with anti-Tbr2 and anti-Ki-67 antibodies, and magnified images of the OSVZ are shown. Arrowheads indicate Tbr2-positive cells co-expressing Ki-67. (N) Percentages of Tbr2-positive cells co-expressing Ki-67 in the OSVZ. (O) Percentages of HOPX-negative oRG cells co-expressing Ki-67 in the OSVZ. n = 3 animals for each condition. Bars present mean ± SD. ns, not significant; Student's *t*-test. Scale bars = 50 µm.

The online version of this article includes the following figure supplement(s) for figure 2:

**Figure supplement 1.** HhipΔC22 electroporation inhibits Shh signaling in the developing ferret cerebral cortex.
**Figure supplement 2.** Distribution of GFP-positive cells in the ferret cerebral cortex.

It remained possible that the HhipΔC22-mediated decrease in the percentage of HOPX-positive oRG cells co-labeled with EdU was due to a decrease in the proliferation of HOPX-positive oRG cells. To exclude this possibility, we examined the effect of HhipΔC22 on cell proliferation. We performed immunostaining for Ki-67 together with HOPX, Tbr2 and Pax6 and found that the percentages of HOPX-positive oRG cells co-expressing Ki-67 were comparable in HhipΔC22-transfected and control cortices (control, 28.7 ± 2.0; HhipΔC22, 29.5 ± 0.7; p=0.30; Student's *t*-test) (*Figure 2K, L*). Similarly, the percentage of IP cells co-expressing Ki-67 and that of HOPX-negative oRG cells co-expressing Ki-67 were not affected by HhipΔC22 (IP cells: control, 36.4 ± 2.8; HhipΔC22, 35.1 ± 3.9; p=0.35; Student's *t*-test) (HOPX-negative oRG cells: control, 28.8 ± 2.7; HhipΔC22, 25.2 ± 3.5; p=0.15; Student's *t*-test) (*Figure 2M–O*). Thus, our findings indicate that Shh signaling suppresses the differentiation of HOXP-positive oRG cells and promotes their self-renewal.

## Activation of Shh signaling is sufficient for increasing HOPX-positive oRG cells

It seemed plausible that the increased self-renewal of HOPX-positive oRG cells in response to Shh signaling leads to an increase in the number of HOPX-positive oRG cells. To test this, we stimulated Shh signaling by introducing Shh-N, an amino-terminal fragment of Shh (*Oberg et al., 2002*; *Sasai et al., 2014*), into the ferret cerebral cortex using IUE at E33. The introduction of Shh-N increased *GLI1* expression in the germinal zone (*Figure 3—figure supplement 1*), indicating that Shh-N efficiently activates Shh signaling in the developing ferret cerebral cortex.

We found that Shh-N markedly increased HOPX-positive oRG cells in GFP-positive transfected areas (*Figure 3A,B*). We then counted the numbers of HOPX-positive oRG cells. To minimize any variation in cell number depending on the positions of coronal sections in the brain, the number of cells on the electroporated side and that on the contralateral non-electroporated side of the cerebral cortex in the same brain section were counted, and the former was divided by the latter (hereafter referred to as the cell number ratio). Our quantification of the cell number ratio showed that HOPX-positive oRG cells were significantly increased by Shh-N (OSVZ: control, 1.00 ± 0.12; Shh-N, 2.38 ± 0.70; p=0.03; Student's *t*-test) (ISVZ: control, 1.05 ± 0.13; Shh-N, 1.71 ± 0.08; p=0.002; Student's *t*-test) (*Figure 3C*), suggesting that the activation of Shh signaling increases HOPX-positive oRG cells in the ferret cerebral cortex. In contrast, the cell number ratios of IP cells were not affected by Shh-N (OSVZ: control, 1.00 ± 0.24; Shh-N, 1.31 ± 0.22; p=0.13; Student's *t*-test) (ISVZ: control, 1.17 ± 0.17; Shh-N, 1.31 ± 0.27; p=0.29; Student's *t*-test) (*Figure 3D,E*). Similarly, triple

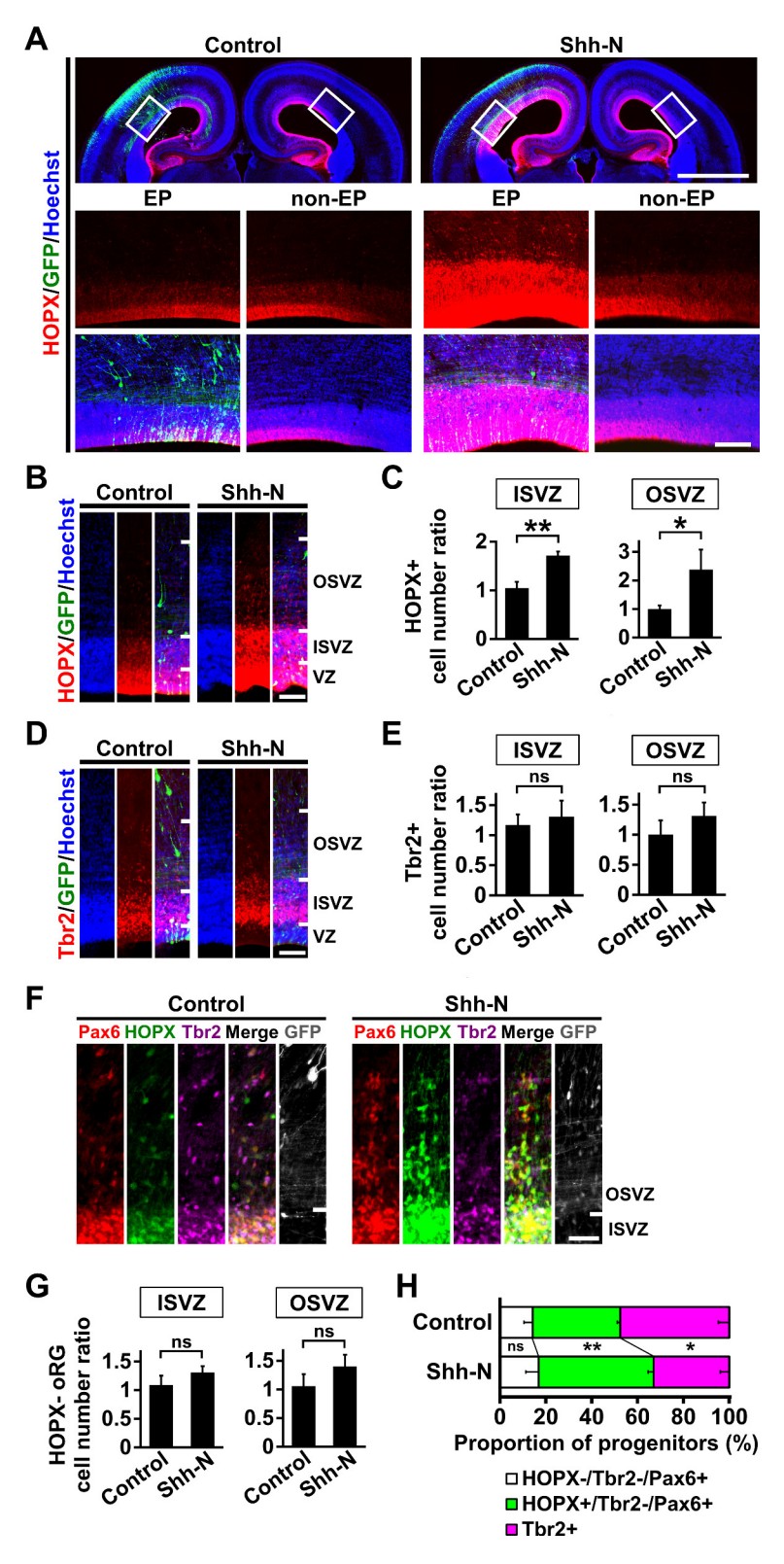

**Figure 3.** Activation of Shh signaling is sufficient to increase HOPX-positive oRG cells in the developing ferret cerebral cortex. pCAG-EGFP plus either pCAG-Shh-N or pCAG control vector was electroporated at E33, and the brains were dissected at P1. (**A**) Coronal sections were stained with anti-HOPX antibody and Hoechst 33342. Boxed areas in the upper panels are magnified in the lower panels. HOPX-positive cells were markedly increased by Shh-N (Shh-N, EP). EP, electroporated side; non-EP, non-electroporated side. Scale bars = 2 mm (upper) and 200 µm (lower). (**B**) Magnified images of

*Figure 3 continued on next page*

Figure 3 continued

the germinal zone. Scale bar = 100 µm. (C) Quantification of HOPX-positive cells. The ratios of the numbers of HOPX-positive cells on the electroporated side relative to those on the non-electroporated side are shown. HOPX-positive cells were significantly increased by Shh-N. n = 3 animals for each condition. Bars present mean ± SD. *p<0.05, **p<0.01, Student's t-test. (D) The germinal zone stained with anti-Tbr2 antibody and Hoechst 33342. Scale bar = 100 µm. (E) Quantification of Tbr2-positive cells. The ratios of the numbers of Tbr2-positive cells on the electroporated side relative to those on the non-electroporated side are shown. n = 3 animals for each condition. Bars present mean ± SD. ns, not significant. Student's t-test. (F) Sections were triple-stained with anti-HOPX, anti-Tbr2 and anti-Pax6 antibodies. Scale bar = 50 µm. (G) Quantification of HOPX-negative oRG cells (Pax6-positive, Tbr2-negative and HOPX-negative). The ratios of the numbers of HOPX-negative oRG cells on the electroporated side relative to those on the non-electroporated side are shown. n = 3 animals for each condition. Bars present mean ± SD. ns, not significant. Student's t-test. (H) Proportions of HOPX-positive oRG cells (green), HOPX-negative oRG cells (white) and IP cells (magenta). HOPX-positive oRG cells were selectively increased by Shh-N. n = 3 animals for each condition. Bars present mean ± SD. ns, not significant. *p<0.05, **p<0.01, Student's t-test.

The online version of this article includes the following figure supplement(s) for figure 3:

**Figure supplement 1.** Shh-N electroporation activates Shh signaling in the ferret cerebral cortex.

immunostaining for HOPX, Pax6 and Tbr2 showed that the cell number ratios of HOPX-negative oRG cells were not affected by Shh-N (OSVZ: control, 1.06 ± 0.21; Shh-N, 1.40 ± 0.21; p=0.09; Student's t-test) (ISVZ: control, 1.09 ± 0.16; Shh-N, 1.31 ± 0.11; p=0.09; Student's t-test) (*Figure 3F,G*).

We next examined the effects of Shh-N on the proportions of HOPX-positive oRG cells, HOPX-negative oRG cells and IP cells (*Figure 3H*). The percentage of HOPX-positive oRG cells was significantly increased by Shh-N (control, 38.3 ± 1.3; Shh-N, 50.2 ± 2.4; p=0.002; Student's t-test) (*Figure 3H*, green). In contrast, the percentages of HOPX-negative oRG cells in Shh-N-transfected and control cortices were comparable (control, 14.2 ± 3.7; Shh-N, 16.9 ± 5.6; p=0.30; Student's t-test) (*Figure 3H*, white), and the percentage of IP cells was smaller in Shh-N-transfected cortices (control, 47.5 ± 4.7; Shh-N, 33.0 ± 3.9; p=0.01; Student's t-test) (*Figure 3H*, magenta). These results indicate that the activation of Shh signaling is sufficient for selectively increasing HOPX-positive oRG cells.

## Shh signaling is essential for producing HOPX-positive oRG cells

We next examined whether endogenous Shh signaling is required for producing HOPX-positive oRG cells by introducing HhipΔC22. We found that HhipΔC22 markedly decreased HOPX-positive oRG cells in EGFP-positive transfected areas (*Figure 4A*, square bracket and *Figure 4B*). Our quantification of the cell number ratio showed that HOPX-positive oRG cells were significantly decreased by HhipΔC22 (OSVZ: control, 1.02 ± 0.07; HhipΔC22, 0.46 ± 0.04; p=0.0003; Student's t-test) (ISVZ: control, 1.03 ± 0.12; HhipΔC22, 0.55 ± 0.04; p=0.003; Student's t-test) (*Figure 4C*).

We then examined the effects of HhipΔC22 on the proportions of HOPX-positive oRG cells, HOPX-negative oRG cells and IP cells (*Figure 4D,E*). Consistent with the decrease in the number of HOPX-positive oRG cells caused by HhipΔC22 (*Figure 4C*), the percentage of HOPX-positive oRG cells was significantly decreased by HhipΔC22 (control, 40.8 ± 2.3; HhipΔC22, 14.5 ± 6.7; p=0.003; Student's t-test) (*Figure 4E*, green). As a result, the percentages of HOPX-negative oRG cells and IP cells were relatively increased by HhipΔC22 (HOPX-negative oRG cells: control, 14.6 ± 1.9; HhipΔC22, 25.1 ± 3.3; p=0.009; Student's t-test) (IP cells: control, 44.6 ± 1.5; HhipΔC22, 60.4 ± 9.4; p=0.039; Student's t-test) (*Figure 4E*, white and magenta). Together, our findings clearly indicate that Shh signaling is required and sufficient for selectively increasing HOPX-positive oRG cells.

We also examined apoptosis using anti-cleaved caspase-3 antibody but did not find significant effects of HhipΔC22 on the cell number ratio of cleaved caspase-3-positive cells in the germinal zone (OSVZ: control, 1.12 ± 0.28; HhipΔC22 0.57 ± 0.49; p=0.12; Student's t-test) (ISVZ: control, 0.97 ± 0.58; HhipΔC22, 0.50 ± 0.41; p=0.20; Student's t-test) (*Figure 4—figure supplement 1*). These results suggest that Shh signaling is irrelevant to apoptosis in the germinal zone of the developing ferret cerebral cortex.

To examine the possibility that Shh signaling promotes the differentiation of HOPX-negative oRG cells into HOPX-positive oRG cells, we quantified the percentage of HOPX-negative oRG cells that started to express *HOPX* mRNA because it has been shown that mRNA levels of marker proteins increase earlier than their protein levels during transient states such as during differentiation (*Berse et al., 2005*; *Lahav et al., 2011*; *Liu et al., 2016*; *Yu et al., 2014*). We performed in situ hybridization for *HOPX* mRNA together with immunostaining for Pax6, Tbr2 and HOPX proteins

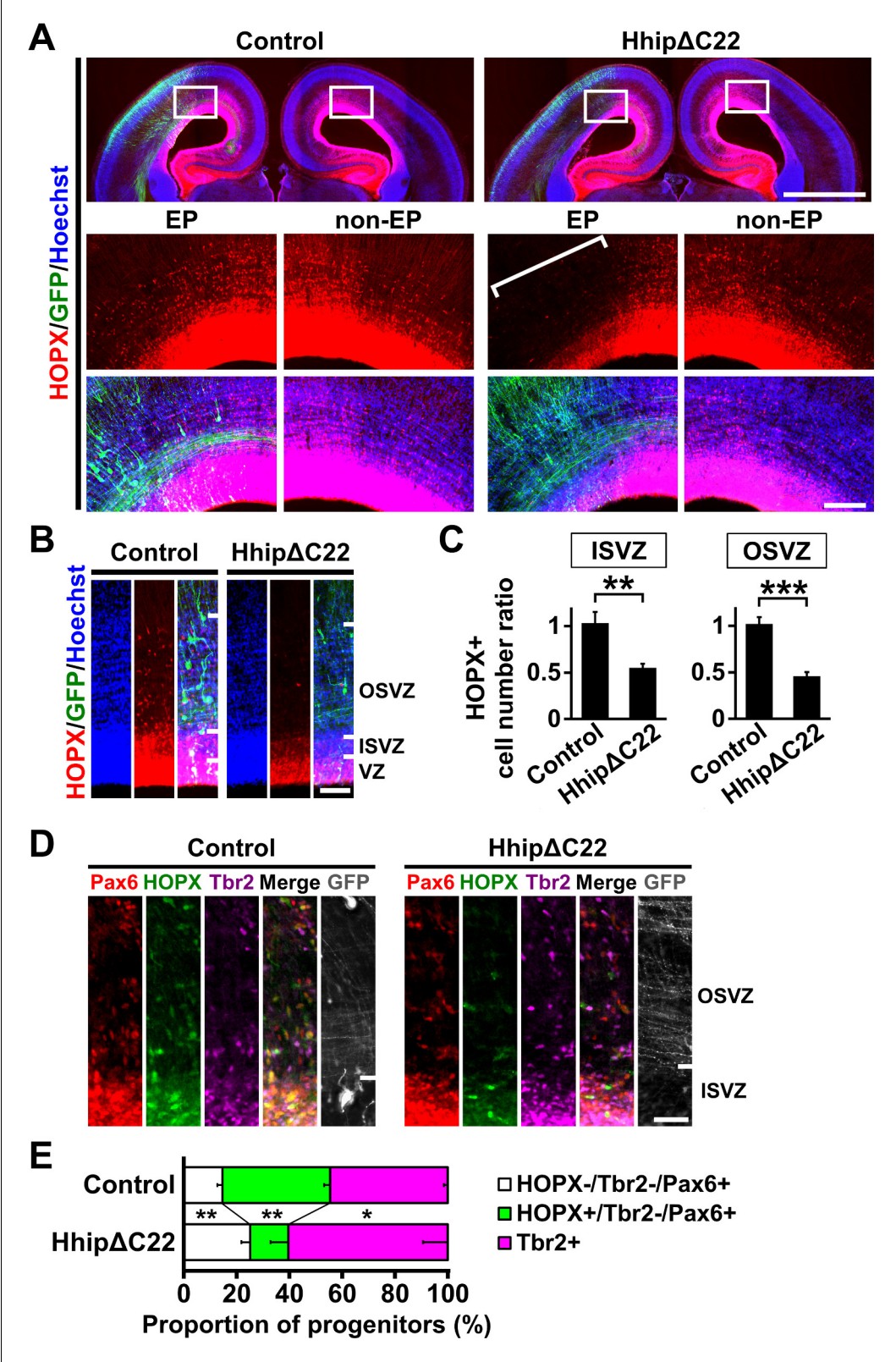

**Figure 4.** Shh signaling is required for producing HOPX-positive oRG cells in the developing ferret cerebral cortex. pCAG-EGFP plus either pCAG-HhipΔC22 or pCAG control vector was electroporated at E33, and the brains were dissected at P1. (**A**) Coronal sections were stained with anti-HOPX antibody and Hoechst 33342. Boxed areas of the upper panels are magnified in the lower panels. HOPX-positive cells were markedly reduced by HhipΔC22 (HhipΔC22, EP, square bracket). EP, electroporated side; non-EP, non-electroporated side. Scale bars = 2 mm (upper panel) and 200 μm

*Figure 4 continued on next page*

*Figure 4 continued*

(lower panel). (B) Magnified images of the germinal zone. Scale bar = 100 µm. (C) Quantification of HOPX-positive cells. The ratios of the numbers of HOPX-positive cells on the electroporated side relative to those on the non-electroporated side are shown. HOPX-positive cells were significantly reduced by HhipΔC22. n = 3 animals for each condition. Bars present mean ± SD. **p<0.01, ***p<0.001, Student's *t*-test. (D) Sections were triple-stained with anti-HOPX, anti-Tbr2 and anti-Pax6 antibodies. Scale bar = 50 µm. (E) Proportions of HOPX-positive oRG cells (green), HOPX-negative oRG cells (white) and IP cells (magenta). HOPX-positive oRG cells were selectively reduced by HhipΔC22. n = 3 animals for each condition. Bars present mean ± SD. *p<0.05, **p<0.01, Student's *t*-test.

The online version of this article includes the following figure supplement(s) for figure 4:

**Figure supplement 1.** Inhibition of Shh signaling does not affect apoptosis in the germinal zone of the developing ferret cerebral cortex.

**Figure supplement 2.** The effects of Shh signaling on the expression of *HOPX* mRNA in the HOPX-negative oRG cells.

(*Figure 4—figure supplement 2A*). We found that the percentage of HOPX-negative oRG cells expressing *HOPX* mRNA were almost comparable between Shh-N-transfected and control cortices (control, 0.9 ± 1.2; Shh-N, 1.1 ± 1.6; p=0.43; Student's *t*-test) (*Figure 4—figure supplement 2B*). Consistently, the percentage of HOPX-negative oRG cells expressing *HOPX* mRNA was not affected by HhipΔC22 (control, 0.6 ± 0.9; HhipΔC22, 0.5 ± 0.7; p=0.45; Student's *t*-test) (*Figure 4—figure supplement 2C*). Although these suggest that Shh signaling is irrelevant to the differentiation of HOPX-negative oRG cells into HOPX-positive oRG cells, it would be important to perform lineage-tracing experiments to confirm this point.

## Shh signaling is necessary and sufficient for cortical folding in the ferret cerebral cortex

Because HOPX-positive oRG cells accumulated in prospective gyral regions, we hypothesized that HOPX-positive oRG cells are crucial for cortical folding. To test this, we increased the number of HOPX-positive oRG cells by introducing Shh-N into the developing ferret cerebral cortex at E33 using IUE and examined cortical folding at P16. We found that Shh-N markedly induced additional gyri and sulci in the electroporated regions, whereas this effect was not observed on the contralateral side or in the GFP-transfected control cortex (*Figure 5A,C*). To quantify this effect of Shh-N, we measured the local gyrification number ratio (local GN ratio) (*Figure 5—figure supplement 1*; *Masuda et al., 2015*). Consistent with our observations, the local GN ratio was significantly increased by Shh-N (control, 0.97 ± 0.05; Shh-N, 1.42 ± 0.06; p=0.003; Student's *t*-test) (*Figure 5B*).

We next examined whether or not Shh-N caused the layer structure of the cerebral cortex to become disorganized. Immunohistochemistry for Brn2, which is highly expressed in layer 2/3, and for Ctip2, which is preferentially expressed in layer 5, demonstrated that layer structures were preserved in the Shh-N-transfected cortex (*Figure 5C*), indicating that additional gyri and sulci in the Shh-N-transfected cortex did not result from the disorganization of cortical layer structures. These results suggest that the activation of Shh signaling is sufficient for inducing additional cortical gyri and sulci in the ferret cerebral cortex.

We next reduced the number of HOPX-positive oRG cells by inhibiting Shh signaling. We found that gyrus size and sulcus depth were markedly reduced on the HhipΔC22-transfected side of the cortex (*Figure 5D*, arrow) compared with the contralateral side of the cortex and the GFP-transfected control cortex (*Figure 5D*, arrowheads). We quantified the local sulcus depth (local SD) ratio (*Figure 5—figure supplement 2*) and the local gyrus size (local GS) ratio (*Figure 5—figure supplement 3*; *Matsumoto et al., 2017b*; *Shinmyo et al., 2017*). The local SD and GS ratios were significantly reduced by HhipΔC22 (local SD ratio: control, 1.03 ± 0.11; HhipΔC22, 0.45 ± 0.03; p=0.006; Student's *t*-test) (local GS ratio: control, 0.86 ± 0.08; HhipΔC22, 0.57 ± 0.06; p=0.038; Student's *t*-test) (*Figure 5E*, local SD ratio and local GS ratio). In contrast, the local GN ratio was not changed by HhipΔC22 (control, 1.06 ± 0.05; HhipΔC22, 1.00 ± 0.00; p=0.19; Student's *t*-test) (*Figure 5E*, local GN ratio). This was presumably because shallow sulci remained on the HhipΔC22-transfected side even though cortical folding was markedly impaired (*Figure 5D*, arrow). Taken together, these results indicate that Shh signaling and HOPX-positive oRG cells are crucial for cortical folding.

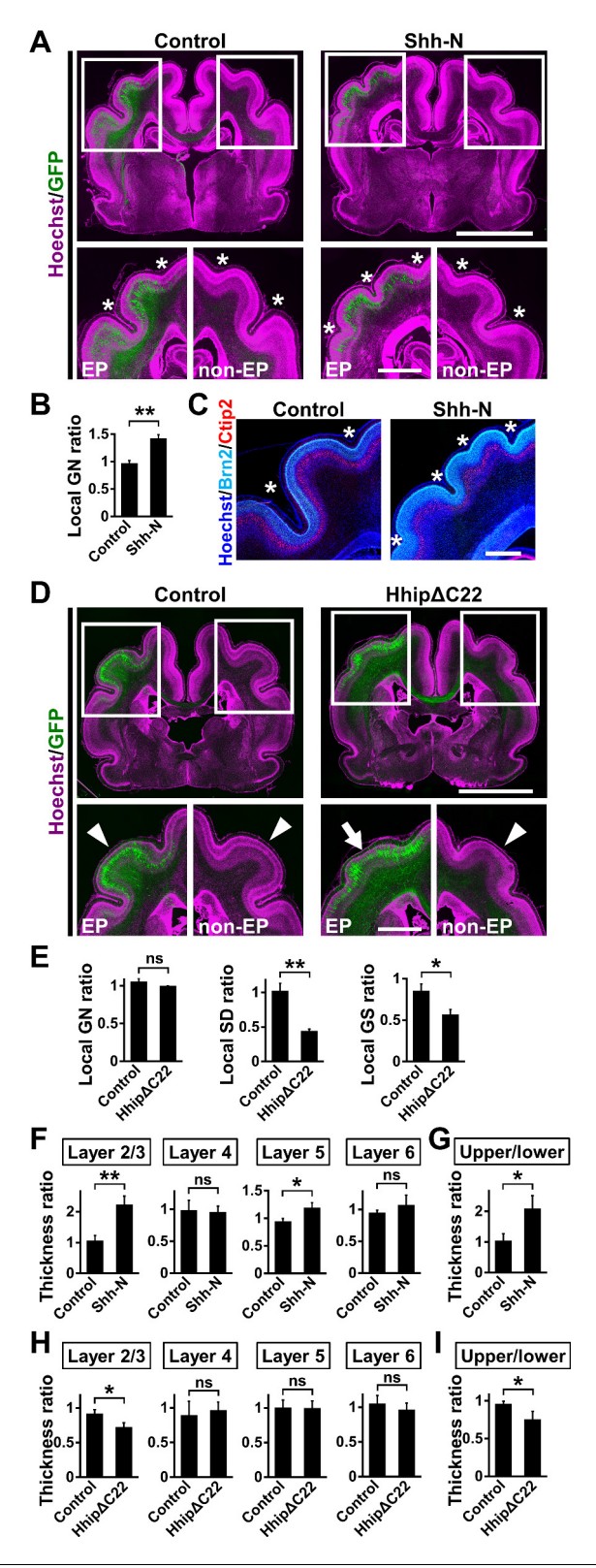

**Figure 5.** Shh signaling is necessary and sufficient for cortical folding in the ferret cerebral cortex. (**A–C**) pCAG-EGFP plus either pCAG-Shh-N or pCAG control vector was electroporated at E33, and the brains were dissected at P16. (**A**) Coronal sections of the electroporated brains were stained with anti-GFP antibody and Hoechst 33342 (magenta). Boxed areas in the upper panels are magnified in the lower panels. Asterisks indicate the positions of
*Figure 5 continued on next page*

*Figure 5 continued*

sulci. The number of sulci was increased in the Shh-N-transfected GFP-positive area (Shh, EP). EP, electroporated side; non-EP, non-electroporated side. Scale bars = 5 mm (upper) and 2 mm (lower). (B) Quantification of the number of cortical folds. The local GN ratio was significantly increased by Shh-N. n = 3 animals for control, n = 4 animals for Shh. Bars present mean ± SEM. **p<0.01, Student's *t*-test. (C) Coronal sections of the electroporated brains were stained with anti-Brn2 (cyan) and anti-Ctip2 (red) antibodies and Hoechst 33342 (blue). Cortical layer structures were preserved in the Shh-N-electroporated area of the cerebral cortex. Asterisks indicate the positions of sulci. Scale bar = 1 mm. (D, E) pCAG-EGFP plus either pCAG-HhipΔC22 or pCAG control vector was electroporated at E33, and the brains were dissected at P16. (D) Coronal sections of the electroporated brains were stained with anti-GFP antibody and Hoechst 33342 (magenta). Cortical folding was markedly suppressed in the HhipΔC22-transfected GFP-positive area (HhipΔC22, EP, arrow). Arrowheads indicate cortical folds in the control cortex and in the non-electroporated side of the cortex. EP, electroporated side; non-EP, non-electroporated side. Scale bars = 5 mm (upper) and 2 mm (lower). (E) Quantification of cortical folding. The local SD ratio and the local GS ratio were significantly smaller in HhipΔC22-transfected samples. n = 3 animals for each condition. Bars present mean ± SEM. ns, not significant. *p<0.05,**p<0.01, Student's *t*-test. (F, G) pCAG-EGFP plus either pCAG-Shh-N or pCAG control vector was electroporated at E33, and the brains were dissected at P16. Coronal sections were subjected to Hoechst 33342 staining plus immunohistochemistry for Ctip2 and Brn2. (F) Quantification of the thicknesses of layer 2/3, layer 4, layer 5 and layer 6. The ratios of the thicknesses of the electroporated side relative to those of the non-electroporated side are shown. Note that the thickness of layer 2/3 was markedly increased by Shh-N. (G) Ratio of upper layer thickness (layer 2/3) to lower layer thickness (layer 4–6). n = 3 animals for each condition. Bars present mean ± SD. *p<0.05, **p<0.01; ns, not significant; Student's *t*-test. (H, I) pCAG-EGFP plus either pCAG-HhipΔC22 or pCAG control vector was electroporated at E33, and the brains were dissected at P16. Coronal sections were subjected to Hoechst 33342 staining plus immunohistochemistry for Ctip2 and Brn2. (H) Quantification of the thicknesses of layer 2/3, layer 4, layer 5 and layer 6. The ratios of the thicknesses of the electroporated side relative to those of the non-electroporated side are shown. Note that the thickness of layer 2/3 was selectively reduced by HhipΔC22. (I) The ratio of upper layer thickness (layer 2/3) to lower layer thickness (layer 4–6). n = 3 animals for each condition. Bars present mean ± SD. *p<0.05; ns, not significant; Student's *t*-test.

The online version of this article includes the following figure supplement(s) for figure 5:

**Figure supplement 1.** Definition of the local gyrification number (GN) ratio.
**Figure supplement 2.** Definition of the local sulcus depth (SD) ratio.
**Figure supplement 3.** Definition of the local gyrus size (GS) ratio.

## Upper layers of the cerebral cortex are preferentially increased by Shh signaling

It has been proposed that the preferential expansion of upper layers mediates the formation of cortical folds (*Kriegstein et al., 2006*; *Richman et al., 1975*). We also previously reported the involvement of the expansion of upper layers in cortical folding (*Masuda et al., 2015*; *Matsumoto et al., 2017b*; *Shinmyo et al., 2017*). Furthermore, a previous report showed that oRG cells predominantly produce upper-layer neurons (*Lukaszewicz et al., 2005*). We therefore hypothesized that increasing HOPX-positive oRG cells by activating Shh signaling results in the preferential expansion of upper layers.

We measured the thickness of each cortical layer and found that Shh-N increased the thickness ratio of layer 2/3 (control, 1.07 ± 0.16; Shh, 2.24 ± 0.27; p=0.003; Student's *t*-test), but not that of layer 4 or layer 6 (layer 4: control, 0.99 ± 0.15; Shh, 0.96 ± 0.09; p=0.42; Student's *t*-test) (layer 6: control, 0.95 ± 0.04; Shh, 1.07 ± 0.15; p=0.17; Student's *t*-test) (*Figure 5F*). Although the thickness ratio of layer 5 was slightly increased by Shh-N (layer 5: control, 0.95 ± 0.05; Shh, 1.19 ± 0.09; p=0.01; Student's *t*-test) (*Figure 5F*), the ratio of layer 2/3 thickness to layer 4–6 thickness was markedly increased by Shh-N (control, 1.04 ± 0.23; Shh, 2.10 ± 0.42; p=0.02; Student's *t*-test) (*Figure 5G*), indicating that the thickness of upper layers is preferentially increased by the activation of Shh signaling.

We also examined the effects of the suppression of Shh signaling on the thickness of each cortical layer. The thickness ratio of layer 2/3 was significantly reduced by HhipΔC22 (control, 0.92 ± 0.06; HhipΔC22, 0.72 ± 0.06; p=0.02; Student's *t*-test), whereas those of layer 4, layer 5 and layer 6 were not affected by HhipΔC22 (layer 4: control, 0.90 ± 0.20; HhipΔC22, 0.97 ± 0.12; p=0.34; Student's *t*-test) (layer 5: control, 1.01 ± 0.11; HhipΔC22, 1.00 ± 0.10; p=0.47; Student's *t*-test) (layer 6: control,

1.05 ± 0.12; HhipΔC22, 0.96 ± 0.10; p=0.22; Student's *t*-test) (*Figure 5H*). The ratio of layer 2/3 thickness to layer 4–6 thickness was significantly decreased by HhipΔC22 (control, 0.96 ± 0.03; HhipΔC22, 0.76 ± 0.10; p=0.03; Student's *t*-test) (*Figure 5I*). It was possible that upper layers were preferentially affected because they were predominantly transfected. However, this seems unlikely because IUE at E33 mainly transfects layer 5 neurons (*Figure 2—figure supplement 2*). These results therefore suggest that Shh signaling preferentially increases upper layer neurons. It seems likely that a preferential increase in upper layers underlies the formation of cortical folds.

## Shh signaling is more activated in the cerebral cortex of ferrets than in that of mice

Our finding that Shh signaling is crucial for cortical folding raised the possibility that Shh signaling is more active in the gyrencephalic ferret brain than in the lissencephalic mouse brain. To test this, we prepared sagittal sections of the developing cerebral cortex of mice and of ferrets and performed in situ hybridization for *GLI1*. As expected, *GLI1* was strongly expressed in the germinal zone of the ferret cerebral cortex (*Figure 6A*), whereas it was barely expressed in the germinal zone of the mouse cerebral cortex (*Figure 6B*). Our quantification showed that the percentage of *GLI1*-positive areas in the germinal zone was significantly higher in ferrets than in mice (ferret, 8.89 ± 1.64%; mouse, 3.69 ± 0.65%; p=0.007; Student's *t*-test) (*Figure 6C*). We also examined the expression levels of *GLI1* mRNA in ferrets at P1 and mice at E17 using RT-PCR (*Figure 6D,E*). Our quantification showed that the relative expression levels of *GLI1* mRNA in the cerebral cortex of ferrets were significantly higher than those of mice (ferret, 0.58 ± 0.16; mouse, 0.30 ± 0.08; p=0.048; Student's *t*-test) (*Figure 6F*). Taken together, these results suggest that Shh signaling is activated in neural progenitors much more in ferrets than in mice.

These results led us to examine whether Shh ligand is more abundant in ferrets than in mice. We prepared protein extracts from cerebral cortices and performed western blotting with anti-Shh antibody. Interestingly, Shh protein levels were much higher in ferrets than in mice (*Figure 6G*). To exclude the possibility that this anti-Shh antibody preferentially recognized ferret Shh compared with mouse Shh, we performed western blotting using cerebella of ferrets and mice. Protein levels of Shh in cerebella were similar in both species (*Figure 6H*), indicating that both ferret Shh and mouse Shh are recognized with similar affinity by this anti-Shh antibody. Our quantification showed that Shh protein levels in the cerebral cortex of ferrets were significantly higher than those in mice (mouse, 0.10 ± 0.01; ferret, 0.32 ± 0.05; p=0.002; Student's *t*-test) (*Figure 6I*). Thus, these results indicate that Shh ligand is more abundant in the cerebral cortex of ferrets than in that of mice. Consistent with this conclusion, *GLI1* mRNA levels in single HOPX-positive cells in the SVZ were significantly higher in ferrets than in mice (mouse, 0.18 ± 0.05; ferret, 0.39 ± 0.06; p=0.04; Student's *t*-test) (*Figure 6J*). It seems plausible that highly activated Shh signaling in the gyrencephalic mammals underlies the expansion of HOPX-positive oRG cells and the acquisition of cortical folds.

## Discussion

We have shown that oRG cells can be classified into two groups: HOPX-positive and HOPX-negative. Shh signaling suppressed the differentiation of HOPX-positive oRG cells, and, as a result, increased the number of HOPX-positive oRG cells. Furthermore, reducing HOPX-positive oRG cells by suppressing Shh signaling impaired cortical folding, while increasing HOPX-positive oRG cells by activating Shh signaling led to additional cortical folds. One plausible scenario is that HOPX-positive oRG cells, which are increased by Shh signaling, preferentially produce upper-layer neurons in prospective gyral regions, resulting in the formation of cortical folds in gyrencephalic brains.

### Subtypes of oRG cells in the developing ferret cerebral cortex

Although the expansion of specific subtypes of SVZ progenitors which are found only in the gyrencephalic brains seemed to be crucial for the formation of cortical folds, the subtypes of SVZ progenitors responsible for cortical folding were unclear. In this study, we revealed that HOPX-positive oRG cells predominantly accumulated in prospective gyral regions compared with HOPX-negative oRG cells. HOPX-positive oRG cells had higher Shh signaling activity and lower differentiation rates than HOPX-negative oRG cells. Furthermore, HOPX-positive oRG cells were selectively increased by the activation of Shh signaling. These several lines of evidence support the idea that HOPX-positive and

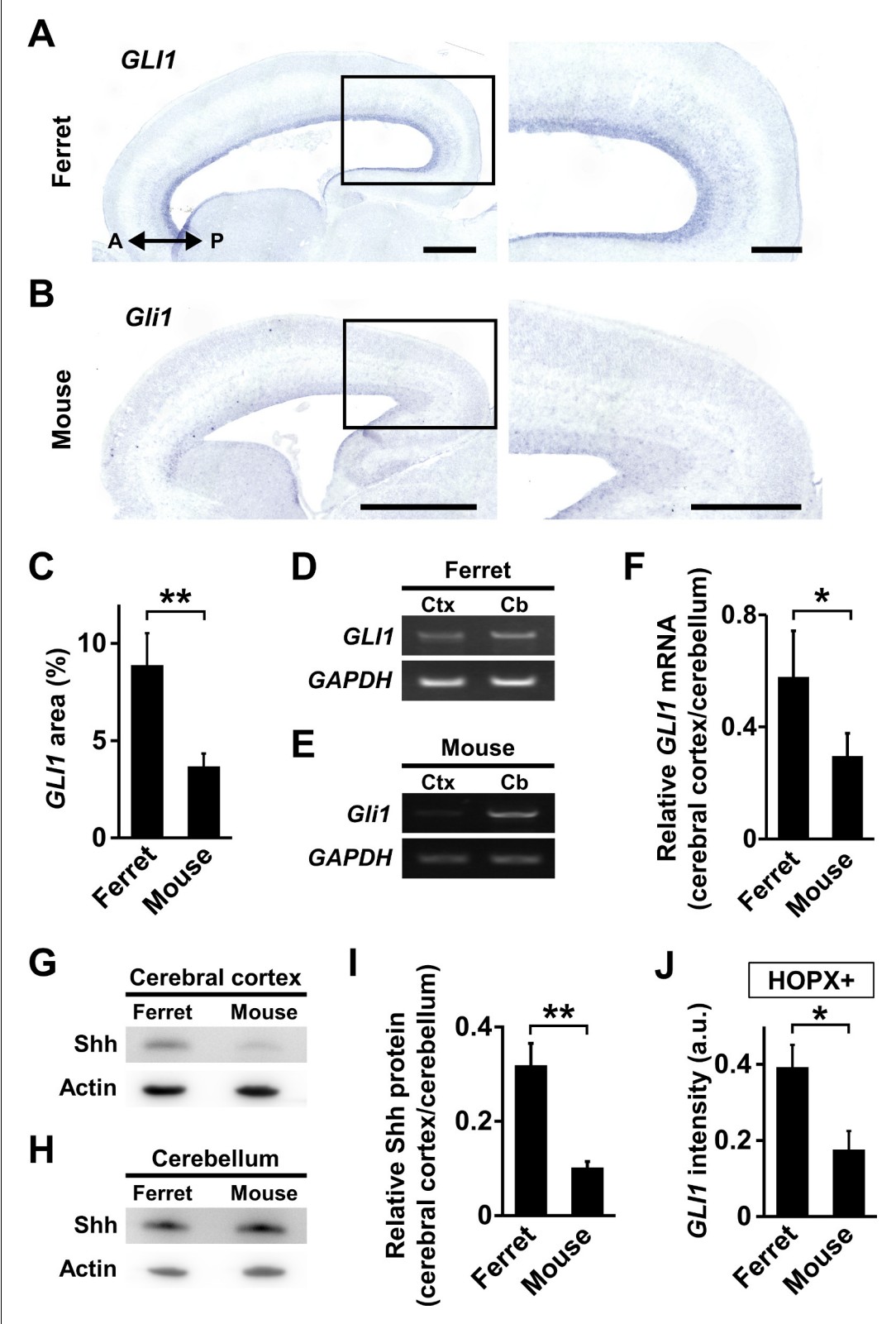

**Figure 6.** Shh signaling is more activated in the cerebral cortex of ferrets than that of mice. (**A, B**) *GLI1* expression in the developing ferret and mouse cerebral cortex. Sagittal sections of the ferret brain at P1 (**A**) and the mouse brain at E17 (**B**) were subjected to in situ hybridization for ferret *GLI1* and mouse *Gli1*, respectively. The areas within the boxes in the left panels are magnified in the right panels. Scale bars = 1 mm (left), 500 μm (right). (**C**) Percentages of *GLI1*-positive areas in the germinal zones of ferret and mouse cerebral cortex. n = 3 animals. Bars present mean ± SD. **p<0.01,

*Figure 6 continued on next page*

*Figure 6 continued*

Student's *t*-test. .(**D**) RT-PCR for *GLI1* and *GAPDH* in the cerebral cortex (Ctx) and the cerebellum (Cb) of ferrets at P1. (**E**) RT-PCR for *Gli1* and *GAPDH* in the cerebral cortex (Ctx) and the cerebellum (Cb) of mice at E17. (**F**) Quantification of *GLI1* mRNA levels in the cerebral cortex of mice and ferrets. *GLI1* signal intensities from RT-PCR were measured, and *GLI1* expression levels in the cerebral cortex were normalized with those in the cerebellum. n = 3 animals. Bars present mean ± SD.*p<0.05, Student's *t*-test. (**G**) Immunoblotting for Shh in the cerebral cortex of mice at E17 and ferrets at P1. (**H**) Immunoblotting for Shh in the cerebellum of mice at E17 and ferrets at P1. (**I**) Quantification of Shh protein levels in the cerebral cortex of mice and ferrets. Shh signal intensities on immunoblots were measured, and Shh signal intensities in the cerebral cortex were normalized with those in the cerebellum. n = 3 animals. Bars present mean ± SD.**p<0.01, Student's *t*-test. (**J**) Quantification of *GLI1* signal intensity in each HOPX-positive cell in the germinal zone of the mouse and ferret cerebral cortex. Sagittal sections of the ferret brain at P1 and the mouse brain at E17 were subjected to in situ hybridization for ferret *GLI1* and mouse *Gli1*, respectively, and to immunohistochemistry for HOPX. The average values of *GLI1* and *Gli1* signal intensities in the germinal zone were normalized with the corresponding values in the external granule layer of the cerebellum in the same sagittal sections. n = 3 animals for each condition. Bars present mean ± SEM.*p<0.05, Student's *t*-test. a.u., arbitrary units.

HOPX-negative oRG cells have distinct cellular properties and that HOPX-positive oRG cells mediate cortical folding. Consistent with this idea, large numbers of HOPX-positive oRG cells were found in the human OSVZ (*Nowakowski et al., 2016*; *Pollen et al., 2015*), while only a small number of HOPX-positive progenitors were found in the mouse germinal zone (*Vaid et al., 2018*). The expansion of HOPX-positive oRG cells seems to be a key feature of gyrencephalic brains. Future investigation is necessary to examine the differences in cellular morphologies, gene expression and cell lineages of HOPX-positive and HOPX-negative oRG cells in gyrencephalic mammals.

In addition to the increased self-renewal of HOPX-positive oRG cells, it seemed possible that the activation of Shh signaling promoted the differentiation of HOPX-negative oRG cells into HOPX-positive oRG cells, resulting in the increase in HOPX-positive oRG cells. However, our data revealed that HOPX-negative oRG cells that started to express *HOPX* mRNA, presumably leading to their differentiation into HOPX-positive oRG cells, were not affected by the activation and inhibition of Shh signaling (*Figure 4—figure supplement 2*). Although these results suggest that differentiation of HOPX-negative oRG cells into HOPX-positive oRG cells would be irrelevant to Shh signaling, it would be important to conduct further experiments to confirm whether HOPX-negative oRG cells differentiate into HOPX-positive oRG cells during development and, if it is the case, whether this differentiation is regulated by Shh signaling or not. It would be important to find a ferret HOPX promoter region which specifically induces transgene expression in HOPX-positive oRG cells for these experiments.

## Mechanisms underlying the differentiation and proliferation of oRG cells

Our previous study demonstrated that the proliferation of oRG cells was regulated by FGF signaling in vivo. When a dominant-negative form of FGF receptor 3 was introduced into the ferret cerebral cortex by IUE, the proliferation of oRG cells was suppressed (*Matsumoto et al., 2017b*). In contrast, the molecular mechanisms responsible for the differentiation of oRG cells remained poorly understood. Our findings clearly indicate that Shh signaling suppresses the differentiation of HOPX-positive oRG cells and promotes their self-renewal in the developing ferret cerebral cortex.

A next intriguing question would be the molecular mechanisms suppressing the differentiation of HOPX-positive oRG cells downstream of Shh signaling. Previously, it was reported that Shh signaling induces the expression of Cdk6 in the developing vertebrate limb (*McGlinn et al., 2005*) and that Cdk6 inhibits the differentiation of myeloid progenitors (*Fujimoto et al., 2007*). Therefore, Shh signaling could suppress the differentiation of oRG cells through Cdk6. Other in vitro studies demonstrated that inhibition of Notch, integrin and STAT signaling resulted in a reduced number of oRG cells (*Fietz et al., 2010*; *Hansen et al., 2010*; *Pollen et al., 2015*). These molecules could also be located downstream of Shh signaling in vivo. To uncover the entire picture of the mechanisms underlying oRG development, it would be important to investigate the roles of these signaling pathways in vivo.

## The role of Shh signaling in cortical folding

Previous pioneering studies using transgenic mice reported that constitutive activation of Shh signaling in neural progenitors induced cortical folds in the medial regions of the cerebral cortex (*Wang et al., 2016*). These transgenic mice, however, did not show obvious cortical folding in most

of the cerebral cortex. Therefore, it was unclear whether Shh signaling was really involved in cortical folding in gyrencephalic mammals. Our gain- and loss-of-function in vivo experiments using ferret IUE clearly indicate that Shh signaling plays a crucial role in cortical folding in the ferret cerebral cortex. Although our results suggest that Shh signaling increases the self-renewal of HOPX-positive oRG cells, resulting in the promotion of cortical folding, it remains possible that Shh signaling has some other effects that promote cortical folding. For example, it was previously reported that Shh activity regulates synaptic connections in the mouse cerebral cortex (*Harwell et al., 2012*). Another recent study suggested that Shh signaling promotes myelination in the developing mouse cerebral cortex (*Zakaria et al., 2019*). These effects of Shh activity could be also involved in cortical folding in the ferret cerebral cortex.

A question that would be intriguing to investigate next is what molecular mechanisms underlie the different responses of HOPX-positive, HOPX-negative oRG cells and IP cells to Shh signaling. One possible mechanism would be that the expression levels of the receptor for Shh, its co-receptors and downstream signaling molecules are different among HOPX-positive, HOPX-negative oRG cells and IP cells. For example, it seems plausible that the expression levels of Patched (Ptch), cell-adhesion-molecule-related/downregulated by oncogenes (Cdon), bioregional Cdon-binding protein (Boc) and growth arrest-specific 1 (GAS1) are different among HOPX-positive and -negative oRG cells and IP cells. It would be important to examine the detailed expression patterns of these molecules in the developing ferret cerebral cortex.

Our results demonstrated that Shh ligand is more abundant in the developing cerebral cortex of ferrets than in that of mice. Furthermore, Shh signaling was more strongly activated in ferrets than in mice. In agreement with this finding, a previous study demonstrated that Shh signaling was more highly activated in the human cerebral cortex than in the mouse cerebral cortex (*Heng et al., 2017*; *Wang et al., 2016*). It is thus conceivable that Shh signaling is commonly activated in the gyrencephalic cerebral cortex of various animal species. It would be therefore intriguing to investigate the mechanisms underlying the increase in Shh ligand in the developing cerebral cortex of gyrencephalic animals (*Yabut and Pleasure, 2018*). High Shh signaling activity seems to be one of the important changes in the evolution of the mammalian brain.

# Materials and methods

**Key resources table**

| Reagent type (species) or resource | Designation | Source or reference | Identifiers | Additional information |
|---|---|---|---|---|
| Antibody | Anti-GFP (rat monoclonal) | Nacalai tesque, Japan | Cat# 440426; RRID:AB_2313652 | IF(1:2000) |
| Antibody | Anti-GFP (rabbit polyclonal) | Medical and Biological Laboratories, Japan | Cat# 598; RRID:AB_591819 | IF(1:2500) |
| Antibody | Anti-Ctip2 (rat monoclonal) | Abcam | Cat# ab18465; RRID:AB_2064130 | IF(1:1000) |
| Antibody | Anti-Brn2 (goat polyclonal) | Santa Cruz Biotechnology | Cat# sc-6029; RRID:AB_2167385 | IF(1:150) |
| Antibody | Anti-Hop (mouse monoclonal) | Santa Cruz Biotechnology | Cat# sc-398703; RRID:AB_2687966 | IF(1:500) |
| Antibody | Anti-Hopx (rabbit polyclonal) | Atlas Antibodies | Cat# HPA030180; RRID:AB_10603770 | IF(1:1000) |
| Antibody | Anti-Tbr2 (sheep polyclonal) | R&D Systems | Cat# AF6166; RRID:AB_ 10569705 | IF(1:100) |
| Antibody | Anti-Tbr2 (rabbit polyclonal) | Abcam | Cat# ab23345; RRID:AB_778267 | IF(1:200) |
| Antibody | Anti-Pax6 (rabbit polyclonal) | Millipore | Cat# AB2237; RRID:AB_1587367 | IF(1:1000) |
| Antibody | Anti-Ki67 (rat monoclonal) | Thermo Fisher Scientific | Cat# 14-5698-80; RRID:AB_10853185 | IF(1:200) |

*Continued on next page*

*Continued*

| Reagent type (species) or resource | Designation | Source or reference | Identifiers | Additional information |
|---|---|---|---|---|
| Antibody | Anti-cleaved caspase 3 (rabbit monoclonal) | BD Pharmingen | Cat# 559565; RRID:AB_397274 | IF(1:300) |
| Antibody | Anti-Shh (rabbit monoclonal) | Cell Signaling Technology | Cat# 2207; RRID:AB_2188191 | WB(1:2500) |
| Antibody | Anti-ß-actin (mouse monoclonal) | Sigma-Aldrich | Cat# A5441; RRID:AB_476744 | WB(1:30000) |
| Antibody | Alkaline phosphatase-conjugated anti-digoxigenin | Roche | Cat# 11093274910; RRID:AB_514497 | ISH(1:2500) |
| Recombinant DNA reagent | pCAG-EGFP (plasmid) | PMID:20181605 | N/A | N/A |
| Recombinant DNA reagent | pCAG control (plasmid) | PMID:26482531 | N/A | N/A |
| Recombinant DNA reagent | pCX-Shh-N (plasmid) | PMID:12112459 | N/A | N/A |
| Recombinant DNA reagent | pCAG-Shh-N (plasmid) | This paper | N/A | N/A |
| Recombinant DNA reagent | pCAGGS-HhipΔC22 (plasmid) | PMID:27558761 | N/A | N/A |
| Recombinant DNA reagent (*Mus musculus*) | pCRII-mouse *Gli1* (plasmid) | This paper (our group) | N/A | vector: pCRII;cDNA fragment: mouse *Gli1*. |
| Recombinant DNA reagent (*Mustela putorius furo*) | pCRII-ferret *GLI1* (plasmid) | This paper (our group) | N/A | vector: pCRII;cDNA fragment: ferret GLI1. |
| Sequence- based reagent (*Mus musculus*) | Mouse *Gli1* forward (primer) | This paper (our group) | N/A | ctctgcttacacagtcagccgcagg |
| Sequence- based reagent (*Mus musculus*) | Mouse *Gli1* reverse (primer) | This paper (our group) | N/A | cccatccctgggcacctcatgtagc |
| Sequence- based reagent (*Mustela putorius fur*) | Ferret *GLI1* forward (primer) | This paper (our group) | N/A | gcatcagctcagcctataccgtc |
| Sequence- based reagent (*Mustela putorius fur*) | Ferret *GLI1* reverse (primer) | This paper (our group) | N/A | tctggctcctcctcccaacttct |
| Sequence- based reagent (*Mustela putorius fur*) | Ferret *HOPX* forward (primer) | This paper (our group) | N/A | ctgtcgccagctctgtaagaggcag |
| Sequence- based reagent (*Mustela putorius fur*) | Ferret *HOPX* reverse (primer) | This paper (our group) | N/A | tcacttggtcttcggccagttggag |
| Sequence- based reagent (*Mustela putorius fur*) | *GAPDH* forward (primer) | This paper (our group) | N/A | gaccacagtccatgccatcact |
| Sequence- based reagent (*Mustela putorius fur*) | *GAPDH* reverse (primer) | This paper (our group) | N/A | tccaccaccctgttgctgtag |
| Commercial assay or kit | Click-iT EdU Alexa Fluor 647 Imaging Kit | Thermo Fisher Scientific | C10340 | N/A |
| Commercial assay or kit | Click-iT Plus Alexa Fluor 488 picolyl Azide Tool Kit | Thermo Fisher Scientific | C10641 | N/A |
| Software, algorithm | FIJI/ImageJ | http://fiji.sc | RRID:SCR_002285 | N/A |
| Software, algorithm | Multi Gauge | Fujifilm | RRID:SCR_014299 | N/A |

## Animals

Normally pigmented sable ferrets (*Mustela putorius furo*) were purchased from Marshall Farms (North Rose, NY). Ferrets were maintained as described previously (*Iwai and Kawasaki, 2009*; *Iwai et al., 2013*; *Kawasaki et al., 2004*).ICR mice were purchased from SLC (Hamamatsu, Japan) and were reared on a normal 12/12 hr light/dark schedule with free access to water and food. The day of conception and that of birth were counted as embryonic day 0 (E0) and postnatal day 0 (P0), respectively. All procedures were performed in accordance with a protocol approved by the Animal Care Committee of Kanazawa University. Randomized mixed-gender cohorts were used for all animal experiments.

## In utero electroporation (IUE) procedure for ferrets

The IUE procedure to express transgenes in the ferret brain was described previously (*Kawasaki et al., 2012*; *Kawasaki et al., 2013*). Briefly, pregnant ferrets were anesthetized, and their body temperature was monitored and maintained using a heating pad. The uterine horns were exposed and kept wet by adding drops of phosphate-buffered saline (PBS) intermittently. The location of embryos was visualized with transmitted light delivered through an optical fiber cable. The pigmented iris was visible, and this enabled us to assume the location of the lateral ventricle. Approximately 2–5 µl of DNA solution was injected into the lateral ventricle at the indicated ages using a pulled glass micropipette. Embryos were randomly selected to electroporate either control plasmid, HhipΔC22-expressing plasmid or Shh-N-expressing plasmid. Each embryo within the uterus was placed between tweezer-type electrodes with a diameter of 5 mm (CUY650-P5; NEPA Gene). Square electric pulses (50–100 V, 50 ms) were passed 5 times at 1 s intervals using an electroporator (ECM830, BTX). Care was taken to quickly place embryos back into the abdominal cavity to avoid excessive temperature loss. The wall and skin of the abdominal cavity were sutured, and the embryos were allowed to develop normally.

Plasmids pCAG-EGFP was described previously (*Sehara et al., 2010*). pCAG-HhipΔC22 was a kind gift from Dr. Daisuke Saito (Kyushu University). Shh-N (an amino-terminal fragment of Shh) was subcloned from pCX-Shh-N (*Oberg et al., 2002*) into pCAG vector, yielding pCAG-Shh-N. Plasmids were purified using the Endofree Plasmid Maxi Kit (Qiagen). For co-transfection, a mixture of pCAG-EGFP plus either pCAG-Shh-N, pCAG-HhipΔC22 or pCAG control plasmid was used. Prior to IUE procedures, plasmid DNA was diluted in PBS, and Fast Green solution was added to a final concentration of 0.5% to monitor the injection.

## Preparation of sections

Sections were prepared as described previously with slight modifications (*Kawasaki et al., 2000*; *Toda et al., 2013*). Briefly, ferrets and mice were deeply anesthetized and transcardially perfused with 4% paraformaldehyde (PFA). The brains were dissected and post-fixed overnight with 4% PFA in PBS. The brains were cryoprotected by 3-day immersion in 30% sucrose and embedded in OCT compound. Sections of 20 or 50 µm thickness were prepared using a cryostat.

## Immunohistochemistry

Immunohistochemistry was performed as described previously with slight modifications (*Kawasaki et al., 2000*; *Toda et al., 2013*). Sections were permeabilized with 0.3% Triton X-100 in PBS and blocked with 2% bovine serum albumin (BSA), 0.3% Triton X-100 in PBS. For quadruple-staining and immunostaining with Ki-67, sections were subjected to microwave antigen retrieval in a sodium citrate solution. The sections were then incubated overnight with primary antibodies, which included anti-Hop (Santa Cruz Biotechnology, RRID:AB_2687966), anti-HOPX (Atlas Antibodies, RRID:AB_10603770), anti-Tbr2 (R&D Systems, RRID:AB_10569705; Abcam, RRID:AB_778267), anti-Pax6 (Millipore, RRID:AB_1587367), anti-Ki-67 (Thermo Fisher Scientific, RRID:AB_10853185), anti-cleaved caspase 3 (BD Pharmingen, RRID:AB_397274), anti-Brn2 (Santa Cruz Biotechnology, RRID:AB_2167385), anti-Ctip2 (Abcam, RRID:AB_2064130), and anti-GFP antibodies (Nacalai Tesque, RRID:AB_2313652; Medical and Biological Laboratories, RRID:AB_591819). After incubation with secondary antibodies and Hoechst 33342, the sections were washed and mounted.

## EdU labeling of ferrets

EdU labeling was performed as described previously with slight modifications (*Turrero García et al., 2016*). 5-Ethynyl-2'-deoxyuridine (EdU) was dissolved in sterile PBS at 1 mg/ml. Newborn ferrets were injected intraperitoneally with EdU (10 mg/kg body weight) at P0, and were sacrificed 28 hr later. To visualize EdU-positive cells, a Click-iT EdU Alexa Fluor Imaging Kit (Molecular Probes) was used according to the manufacturer's instructions after immunostaining was performed as described above.

## In situ hybridization

In situ hybridization was performed as described previously (*Matsumoto et al., 2017a*; *Matsumoto et al., 2017b*). Sections prepared from fixed tissues were treated with 4% PFA for 10 min, 1 µg/ml proteinase K for 10 min and 0.25% acetic anhydride for 10 min. After prehybridization, the sections were incubated overnight at 58 ˚C with digoxigenin-labeled RNA probes diluted in hybridization buffer (50% formamide, 5x SSC, 5x Denhardt's solution, 0.3 mg/ml yeast RNA, 0.1 mg/ml herring sperm DNA and 1 mM dithiothreitol). The sections were then incubated with alkaline phosphatase-conjugated anti-digoxigenin antibody (Roche, RRID:AB_514497) and Hoechst 33342, and visualized using NBT/BCIP as a substrate.

For combined in situ hybridization and immunostaining, sections were incubated with anti-Hop, anti-Tbr2 and anti-Pax6 antibodies in 2% BSA, 0.02% DEPC and 0.3% Triton X-100/PBS. After incubation with secondary antibodies, images of the sections were acquired using a BZ-9000 microscope (Keyence). After hybridization was performed, the sections were incubated with anti-GFP (Medical and Biological Laboratories, RRID:AB_591819) and alkaline phosphatase-conjugated anti-digoxigenin antibodies. After incubation with secondary antibodies, in situ signals were visualized with NBT/BCIP. Probes used here were made as follows. Ferret *HOPX*, ferret *GLI1* and mouse *Gli1* cDNA fragments were amplified using RT-PCR and inserted into the pCRII vector. The sequences of primers used to amplify ferret *HOPX*, ferret *GLI1* and mouse *Gli1* cDNA fragments were as follows: ferret *HOPX*, forward ctgtcgccagctctgtaagaggcag, reverse tcacttggtcttcggccagttggag; ferret *GLI1*, forward gcatcagctcagcctataccgtc, reverse tctggctcctcctcccaacttct; mouse *Gli1*, forward ctctgcttacacagtcagccgcagg, reverse cccatccctgggcacctcatgtagc. PCR products made using these primers were confirmed by DNA sequencing.

## Immunoblotting

Immunoblotting was performed as described previously with slight modifications (*Kawasaki et al., 1999*). In brief, tissue lysates were separated by SDS-PAGE and transferred onto a PVDF membrane. After blocking with 5% skim milk, the membrane was then incubated overnight with primary antibodies, which included anti-Shh (Cell Signaling Technology, RRID:AB_2188191) and anti-ß-actin (Sigma-Aldrich, RRID:AB_476744) antibodies, and subsequently with horseradish peroxidase-conjugated secondary antibodies. Signals were detected by the ECL Plus Western blotting detection system (GE Healthcare Life Science). Signal intensities of bands were measured using Multi Gauge software (Fujifilm).

## RT-PCR

RT-PCR was performed as described previously with modifications (*Kawasaki et al., 2002*). Total RNA was isolated from the cerebral cortex and the cerebellum of P1 ferrets and E17 mice using the RNeasy Mini Kit (Qiagen) and treated with DNase I to eliminate genomic DNA contamination. Reverse transcription was performed using oligo(dT)12–18 (Thermo Fisher Scientific) and Superscript III (Thermo Fisher Scientific). Samples without Superscript III were also made as negative controls. PCR was performed using the following primers: ferret *GLI1* forward, gcatcagctcagcctataccgtc; ferret *GLI1* reverse, tctggctcctcctcccaacttct; mouse *Gli1* forward, ctctgcttacacagtcagccgcagg; mouse *Gli1* reverse, cccatccctgggcacctcatgtagc; *GAPDH* forward, gaccacagtccatgccatcact; *GAPDH* reverse, tccaccaccctgttgctgtag. PCR products made using these primers were confirmed by DNA sequencing. Signal intensities of bands were measured using the 'ROI manager' tool of ImageJ software. Relative expression levels were calculated using the following formula:

Relative *GLI1* expression = (cerebral cortex *GLI1*/cerebral cortex *GAPDH*) / (cerebellum *GLI1*/cerebellum *GAPDH*)

The expression levels in the cerebral cortex were normalized with those in the cerebellum because the affinities of the primers could be different between ferrets and mice.

## Cell counting

The numbers of immunopositive cells in a column with 100–600 μm width were manually counted using the 'cell counter' tool of ImageJ software. Threshold values of signals were measured using the ImageJ 'threshold' tool, and cells that had signals stronger than the threshold values were counted. To minimize variation of cell numbers depending on the positions of coronal sections in the brain, the number of cells in a column from the electroporated side and the number in a column from the contralateral non-electroporated side of the cerebral cortex in the same brain section were counted, and the former was divided by the latter (cell number ratio). The cell-dense layer next to the VZ was identified as the ISVZ, and the cell-sparse layer between the ISVZ and the IZ was identified as the OSVZ.

## Calculation of the local GN ratio, the local GS ratio and the local SD ratio

Calculation of the local GN, GS and SD ratios was performed as described previously with slight modifications (*Masuda et al., 2015*; *Matsumoto et al., 2017b*). Serial coronal sections containing the suprasylvian gyrus (SSG) were prepared. Sections were taken every 500 μm from the more posterior part of the ansate sulcus. The selected sections containing the SSG were stained with Hoechst 33342, and images of whole sections were acquired using a BZ-9000 microscope (Keyence). The averages of the local GS ratios and the local SD ratios were calculated using three sections for each animal, and those of the local GN ratios were calculated using six sections for each animal.

To calculate the local GN ratio, we counted how many times the complete contour (*Figure 5— figure supplement 1*, green line) was detached from the outer contour (*Figure 5—figure supplement 1*, red line) of the cerebral cortex (local GN value). To minimize variation of the local GN values depending on the positions of coronal sections in the brain, the local GN value on the electroporated side was divided by that on the contralateral non-electroporated side of the cerebral cortex in the same brain section (local GN ratio). The local GN ratio would be one if the number of sulci was the same between the electroporated side and the non-electroporated side, and would be higher than one if cortical folding was increased by genetic manipulation.

To calculate the local SD ratio, a line connecting the top of the SSG and the top of the lateral gyrus was drawn (*Figure 5—figure supplement 2*, red line). A green line connecting the bottom of the lateral sulcus (LS) and the red line was then drawn so that the green line was perpendicular to the red line (*Figure 5—figure supplement 2*). The length of the green line was used as the local SD value. To minimize variation of the local SD values depending on the positions of coronal sections in the brain, the local SD value on the electroporated side was divided by that on the contralateral non-electroporated side of the cerebral cortex in the same brain section (local SD ratio). The local SD ratio would be one if the depth of the LS was the same between the electroporated side and the non-electroporated side, and would be smaller than one if the depth of the LS was reduced by genetic manipulation. In some cases, the suprasylvian sulcus (SSS) was used instead of the LS.

To calculate the local GS ratio, the area surrounded by the brain surface (*Figure 5—figure supplement 3*, green line) and the red line connecting the bottom of the SSS and that of the LS was measured (local GS value) (*Figure 5—figure supplement 3*). To minimize variation of the local GS values depending on the positions of coronal sections in the brain, the local GS value on the electroporated side was divided by that on the contralateral non-electroporated side of the cerebral cortex in the same brain section (local GS ratio). The local GS ratio would be one if the size of SSG was the same between the electroporated side and the non-electroporated side, and would be smaller than one if cortical folding was suppressed by genetic manipulation.

## Quantification of the thickness of cortical layers

Cortical layer thickness was quantified as described previously with slight modifications (*Matsumoto et al., 2017b*). Coronal sections containing the SSG and the MEG were used. The sections were subjected to Hoechst 33342 staining plus immunohistochemistry for Ctip2 and Brn2, and images of the sections were acquired using a BZ-9000 microscope (Keyence). Layer 2/3 and layer 4

were identified using Hoechst and Brn2 images, and layer 5 and layer 6 were identified using Hoechst and Ctip2 images. To calculate the thickness of each cortical layer, the area of each layer was measured and divided by the tangential length of the area (thickness value). To minimize variation of the thickness value depending on the positions of coronal sections in the brain, the thickness value on the electroporated side was divided by that on the contralateral non-electroporated side of the cerebral cortex in the same brain section (thickness ratio). The thickness ratio would be one if the thickness was the same between the electroporated side and the non-electroporated side, and would be smaller than one if the thickness was reduced by genetic manipulation.

## Quantification of the *GLI1*-positive area

Sagittal sections of mouse and ferret brains were prepared at E17 and P1, respectively. The sections were subjected to Hoechst 33342 staining and in situ hybridization for mouse *Gli1* or ferret *GLI1*. Images of the sections were acquired using a BZ-9000 microscope (Keyence). The borders of the germinal zone were determined using Hoechst images. After background signals were removed using the 'threshold' tool of ImageJ, the *GLI1*-positive area was measured in a column with 300 µm width within the germinal zone of the cerebral cortex. The total area of the column within the germinal zone was also measured. The *GLI1*-positive area was divided by the total area to normalize the size of the germinal zone between mice and ferrets.

## Quantification of *GLI1* signal intensity

Sagittal sections of medial parts of the cerebral cortex were prepared from E17 mice and P1 ferrets. The sections were subjected to immunostaining for HOPX, and images of the sections were acquired using a BZ-9000 microscope (Keyence). The sections were then subjected to Hoechst 33342 staining and in situ hybridization for ferret *GLI1* or mouse *Gli1*. *GLI1* signal intensity in each HOPX-positive cell was measured and used for calculating the average value of *GLI1* signal intensity in each animal. The average value was divided by *GLI1* signal intensity in the external granule layer of the cerebellum in the same sagittal section for normalization.

## Acknowledgements

We thank Drs. Shigetada Nakanishi (Suntory Foundation for Life Science) and Charles Yokoyama (The University of Tokyo) for their critical reading of this manuscript, and Drs. Noriyuki Sasai (Nara Institute of Science and Technology) and Daisuke Saito (Kyushu University) for plasmids. We are grateful to Zachary Blalock, Miwako Hirota, Natsu Uda and Kawasaki lab members for their helpful support.

## Additional information

### Funding

| Funder | Author |
| --- | --- |
| Ministry of Education, Culture, Sports, Science, and Technology | Hiroshi Kawasaki<br>Yohei Shinmyo<br>Naoyuki Matsumoto |
| Japan Agency for Medical Research and Development | Hiroshi Kawasaki |
| Takeda Science Foundation | Naoyuki Matsumoto<br>Yohei Shinmyo<br>Hiroshi Kawasaki |
| Mitsubishi Foundation | Hiroshi Kawasaki |
| Uehara Memorial Foundation | Yohei Shinmyo |
| Mochida Memorial Foundation for Medical and Pharmaceutical Research | Yohei Shinmyo |
| Hokuriku Bank Research Grant for Young Scientist | Naoyuki Matsumoto |

| Kanazawa University Sakigake Project | Hiroshi Kawasaki |
| Kanazawa University Chozen Project | Hiroshi Kawasaki |

The funders had no role in study design, data collection and interpretation, or the decision to submit the work for publication.

## Author contributions
Naoyuki Matsumoto, Conceptualization, Resources, Validation, Investigation, Methodology, Writing - original draft, Writing - review and editing; Satoshi Tanaka, Toshihide Horiike, Validation, Investigation, Methodology; Yohei Shinmyo, Validation, Investigation, Methodology, Writing - review and editing; Hiroshi Kawasaki, Supervision, Funding acquisition, Writing - original draft, Project administration, Writing - review and editing

## Author ORCIDs
Hiroshi Kawasaki (iD) https://orcid.org/0000-0002-2514-1497

## Ethics
Animal experimentation: All procedures were performed in accordance with a protocol approved by the Animal Care Committee of Kanazawa University.

## Decision letter and Author response
Decision letter https://doi.org/10.7554/eLife.54873.sa1
Author response https://doi.org/10.7554/eLife.54873.sa2

# Additional files

## Supplementary files
• Supplementary file 1. Data sets for quantification.
• Transparent reporting form

## Data availability
All data generated or analysed during this study are included in the manuscript and supporting files.

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
