## [Decision Letter]

**Acceptance summary:**

This study provides important insights into the mechanisms promoting cortical folding in the ferret. The authors show that expression of the HOPX protein distinguishes outer radial glial cells with a high self-renewal activity that are located in prospective gyral regions of the cortex. They further demonstrate that Sonic hedgehog signaling promotes the expansion of the HOPX-positive progenitor population and stimulates cortical folding.

**Decision letter after peer review:**

Thank you for submitting your article "A discrete subtype of neural progenitor crucial for cortical folding in the gyrencephalic mammalian brain" for consideration by *eLife*. Your article has been reviewed by three peer reviewers, and the evaluation has been overseen by Francois Guillemot as Reviewing Editor and Marianne Bronner as the Senior Editor. The following individuals involved in review of your submission have agreed to reveal their identity: Frank Jacobs (Reviewer #2); Kinichi Nakashima (Reviewer #3).

The reviewers have discussed the reviews with one another and the Reviewing Editor has drafted this decision to help you prepare a revised submission.

Summary:

This is a well performed study reporting that in the ferret cortex, outer radial glial progenitors (oRGs) that express the homeobox transcription factor HOPX contribute to gyrification while HOPX-negative oRGs do not contribute. They further demonstrate that HOPX+ oRGs have a high self-renewing activity and that their numbers are increased by Sonic Hedgehog signaling activity. Finally, they provide evidence that Shh is activated at a higher level in the gyrencephalic ferret cortex than in the lissencephalic murine cortex.

Overall this study provides strong evidence by loss of function analysis in the ferret cortex of a role of Shh in cortical gyrification, and thus provides important support for a model put forward by an earlier study based on a gain of function study in the mouse. The distinction between HOPX+ and HOPX- oRGs is also interesting, although the relationship between the two cell populations needs to be further address as explained below.

Essential revisions:

1) Figure 1 and 2: the ferret cortices were analyzed on P1 and it remains unclear whether HOPX-negative cells are a truly separate subtype of oRG cells, or if they merely represent a transient precursor state for HOPX-positive oRG cells, in which case Shh might drive the generation of HOPX+ oRGs from HOPX- cells. A lineage tracing experiment does not seem feasible within the two months period of revision, but the authors should nevertheless address experimentally the possibility that HOPX+ cells originate form HOPX- cells, e.g. by quantifying the two cell populations and their proliferation after 24hr exposure to Shh-N and inhibitor. They should also discuss in the text the possibility that such a lineage relationship exists.

2) Figure 1C/G: The authors suggest that there is a reduction of HOPX-positive cells in prospective sulci regions outside of the prospective gyri. This dataset is presented as an important support for the paper's main conclusions about the involvement of HOPX-positive cells in cortical folding. To validate the observation, the authors quantify the ratio of HOPX-positive cells (and other types of progenitors) between region #2 and the most distal region #5. The author's hypothesis would be even more strongly supported if the quantitative analysis included the adjacent prospective gyral regions #1 and #3 as cells in these more proximal regions are less likely to differ in regional identity.

3) Figure 1J and K: the authors write "HOPX-positive oRG cells tend to remain in their progenitor state even after cell division and do not differentiate into the next stage". However, to support this interpretation, they need to perform immunohistochemistry using anti-Ki67 antibody together with EdU staining 28h after EdU injection to show that many EdU-positive cells are Ki67-positive and are therefore still in proliferative state. Alternatively, they could perform chase labeling with another dT analogue, such as CdU and IdU, 28h after EdU injection and show that many cells are dual positive for the two dT analogues.

4) Figure 2JK. The authors show that the percentage of HOPX-positive oRGs co-labeled with Edu was diminished by Shh inhibition. However an important control, an analysis of Edu incorporation in HOPX-negative oRGs, is missing. The TBR2/Edu ratio presented in Figure 2H/I is not as good a control as TBR2 marks a different type of progenitors.

5) Figure 3: The increase in HOPX expression by ectopic Shh-N expression seems to be accompanied by an increase of thickness of the germinal layers (both VZ and iSVZ; Figure 3A/B), and the increase of HOPX-positive progenitors does not seem dramatically higher than the increase of the HOPX-negative population. To make their argument stronger, in addition to the relative increase in HOPX-positive cells in the ISVZ and OSVZ in Panel C, the authors should perform a similar calculation of the increases of other type of progenitors.

6) Regarding the laminar specificity of the effect of ectopic Shh activation discussed in the subsection “Upper layers of the cerebral cortex are preferentially increased by Shh signaling”: The authors should clarify whether the timepoint of Shh-N transfection (E33) is not too late to affect cortical layer VI and V neurons. Are the observed effects of Shh activation only observed in upper cortical layer neurons because these are the only ones being generated after E33? If this is the case, the authors should perform IUE earlier to assess the effect of Shh-N transfection during the earlier waves of neurogenesis that form cortical layer VI and/or layer V neurons.

7) The authors write "As expected, GLI1 was strongly expressed in the germinal zone of the ferret cerebral cortex (Figure 6A), whereas it was barely expressed in the germinal zone of the mouse cerebral cortex (Figure 6B)." However, since the probes against ferret and mouse Gli mRNA are different, it is not possible to directly compare expression levels of Gli mRNA in the brain section from different species. The authors should perform RT-PCR for Gli mRNA and measure the relative amount compared to housekeeping genes.

---

## [Author Response]

Essential revisions:1) Figure 1 and 2: the ferret cortices were analyzed on P1 and it remains unclear whether HOPX-negative cells are a truly separate subtype of oRG cells, or if they merely represent a transient precursor state for HOPX-positive oRG cells, in which case Shh might drive the generation of HOPX+ oRGs from HOPX- cells. A lineage tracing experiment does not seem feasible within the two months period of revision, but the authors should nevertheless address experimentally the possibility that HOPX+ cells originate form HOPX- cells, e.g. by quantifying the two cell populations and their proliferation after 24hr exposure to Shh-N and inhibitor. They should also discuss in the text the possibility that such a lineage relationship exists.

It has been shown that mRNA levels of marker proteins increase earlier than their protein levels during transient states such as differentiation (Liu et al., 2016; Berse et al., 2005; Lahav et al., 2011; Yu et al., 2014). Therefore, in accordance with the reviewer's comment, we investigated whether the activation of Shh signaling promotes differentiation of HOPX-negative oRG cells into HOPX-positive oRG cells. We newly made an RNA probe for ferret *HOPX*, and by combining in situ hybridization for *HOPX* mRNA with immunostaining for HOPX, Tbr2 and Pax6, we quantified the percentage of HOPX-negative oRG cells that express *HOPX* mRNA. Presumably, these cells had started to express *HOPX* mRNA and would subsequently differentiate into HOPX-positive oRG cells. We found that the percentages of HOPX-negative oRG cells expressing *HOPX* mRNA were almost comparable among HhipΔC22-transfected, Shh-N-transfected and control cortices. Thus, it seems unlikely that Shh signaling promotes the differentiation of HOPX-negative oRG cells into HOPX-positive oRG cells. These results were added as new Figure 4—figure supplement 2, and were written in the Results section (subsection “Shh signaling is essential for producing HOPX-positive oRG cells”, last paragraph). In addition, as suggested, we also added a discussion of the potential lineage relationship between HOPX-positive and HOPX-negative progenitors in the Discussion section (subsection “Subtypes of oRG cells in the developing ferret cerebral cortex”, last paragraph).

2) Figure 1C/G: The authors suggest that there is a reduction of HOPX-positive cells in prospective sulci regions outside of the prospective gyri. This dataset is presented as an important support for the paper's main conclusions about the involvement of HOPX-positive cells in cortical folding. To validate the observation, the authors quantify the ratio of HOPX-positive cells (and other types of progenitors) between region #2 and the most distal region #5. The author's hypothesis would be even more strongly supported if the quantitative analysis included the adjacent prospective gyral regions #1 and #3 as cells in these more proximal regions are less likely to differ in regional identity.

In accordance with the reviewer's comment, we quantified the ratios of the numbers of progenitor cells and the proportions of progenitor cells in adjacent prospective gyral and sulcal regions #2 and #3. Consistent with our original results using region #2 and #5, we found that HOPX-positive oRG cells, rather than HOPX-negative oRG cells and IP cells, preferentially accumulated in the prospective gyral regions. We replaced our original quantification using regions #2 and #5 (original Figure 1F and G) with new quantification using regions #2 and #3 (new Figure 1F and G). Our original quantification was moved to Figure 1—figure supplement 1. These results were written in the text (subsection “HOPX-positive and HOPX-negative oRG cells in the developing ferret cerebral 112 cortex have distinct cellular properties”).

3) Figure 1J and K: the authors write "HOPX-positive oRG cells tend to remain in their progenitor state even after cell division and do not differentiate into the next stage". However, to support this interpretation, they need to perform immunohistochemistry using anti-Ki67 antibody together with EdU staining 28h after EdU injection to show that many EdU-positive cells are Ki67-positive and are therefore still in proliferative state. Alternatively, they could perform chase labeling with another dT analogue, such as CdU and IdU, 28h after EdU injection and show that many cells are dual positive for the two dT analogues.

In accordance with the reviewer's comment, we performed immunohistochemistry for Ki-67 and EdU staining 28 hrs after EdU injection. We then quantified the percentage of HOPX-positive oRG cells which were also positive for Ki-67 and EdU, and the percentage of HOPX-negative oRG cells which were also positive for Ki-67 and EdU. Consistent with our idea, we found that the percentage of HOPX-positive oRG cells co-labeled with EdU and Ki-67 was significantly higher than that of HOPX-negative oRG cells co-labeled with EdU and Ki-67, indicating that HOPX-positive oRG cells tend to remain in their proliferative state even after cell division. This result was added as new Figure 1L and M, and was written in the Results section (subsection “HOPX-positive and HOPX-negative oRG cells in the developing ferret cerebral cortex have distinct cellular properties”, last paragraph).

4) Figure 2J, K. The authors show that the percentage of HOPX-positive oRGs co-labeled with Edu was diminished by Shh inhibition. However an important control, an analysis of Edu incorporation in HOPX-negative oRGs, is missing. The TBR2/Edu ratio presented in Figure 2H/I is not as good a control as TBR2 marks a different type of progenitors.

As suggested by the reviewer, we quantified the percentage of HOPX-negative oRG cells co-labeled with EdU in HhipΔC22-transfected and control cortices, and found that the percentage was not affected by HhipΔC22. In addition, we also quantified the percentage of HOPX-negative oRG cells co-expressing Ki-67 in HhipΔC22-transfected and control cortices, and found that the percentage was also not affected by HhipΔC22. These results are consistent with idea that Shh signaling selectively regulates the self-renewal of HOXP-positive oRG cells in the ferret cerebral cortex. These results were added as new Figures 2J and O, and were written in the Results section (subsection “Shh signaling enhances the self-renewal of HOPX-positive oRG cells and suppresses their differentiation”, last paragraph).

5) Figure 3: The increase in HOPX expression by ectopic Shh-N expression seems to be accompanied by an increase of thickness of the germinal layers (both VZ and iSVZ; Figure 3A/B), and the increase of HOPX-positive progenitors does not seem dramatically higher than the increase of the HOPX-negative population. To make their argument stronger, in addition to the relative increase in HOPX-positive cells in the ISVZ and OSVZ in Panel C, the authors should perform a similar calculation of the increases of other type of progenitors.

In accordance with the reviewer's comment, we examined the cell number ratios of IP cells and HOPX-negative oRG cells in Shh-N-transfected and control cortices. We found that the cell number ratios of IP cells and HOPX-negative oRG cells were comparable between Shh-N-transfected and control cortices. These results support our idea that HOPX-positive oRG cells are preferentially regulated by Shh signaling. These results were added as new Figures 3D, E and G, and were written in the Results section (subsection “Activation of Shh signaling is sufficient for increasing HOPX-positive oRG cells”, second paragraph).

6) Regarding the laminar specificity of the effect of ectopic Shh activation discussed in the subsection “Upper layers of the cerebral cortex are preferentially increased by Shh signaling”: The authors should clarify whether the timepoint of Shh-N transfection (E33) is not too late to affect cortical layer VI and V neurons. Are the observed effects of Shh activation only observed in upper cortical layer neurons because these are the only ones being generated after E33? If this is the case, the authors should perform IUE earlier to assess the effect of Shh-N transfection during the earlier waves of neurogenesis that form cortical layer VI and/or layer V neurons.

Importantly, electroporation at E33 mainly transfects layer 5 neurons in the ferret cerebral cortex. However, our results showed that the thicknesses of layers 4 and 5 were less affected by electroporation of Shh-N and HhipΔC22 at E33 compared with those of layer 2/3. It therefore seemed unlikely that the preferential changes in layer 2/3 resulted from the distribution of transfected cells. We added new Figure 2—figure supplement 2, which shows the distribution of GFP-positive cells in the ferret cerebral cortex electroporated at E33 and wrote this information in the text (subsection “Shh signaling enhances the self-renewal of HOPX-positive oRG cells and suppresses their differentiation”, second paragraph; subsection “Upper layers of the cerebral cortex are preferentially increased by Shh signaling”, last paragraph).

7) The authors write "As expected, GLI1 was strongly expressed in the germinal zone of the ferret cerebral cortex (Figure 6A), whereas it was barely expressed in the germinal zone of the mouse cerebral cortex (Figure 6B)." However, since the probes against ferret and mouse Gli mRNA are different, it is not possible to directly compare expression levels of Gli mRNA in the brain section from different species. The authors should perform RT-PCR for Gli mRNA and measure the relative amount compared to housekeeping genes.

In accordance with the reviewer's comment, we performed RT-PCR and compared the expression levels of *GLI1* mRNA between mice and ferrets. We found that relative expression levels of *GLI1* mRNA in the cerebral cortex of ferrets were higher than those in mice. Taken together with our results using in situ hybridization, our results indicate that Shh signaling is more active in the cerebral cortex of ferrets than in that of mice. These results were added as new Figures 6D, E and F, and was written in the text (subsection “Shh signaling is more activated in the cerebral cortex of ferrets than in that of mice”, first paragraph).